# Secretin-dependent signals in the ventromedial hypothalamus regulate energy metabolism and bone homeostasis in mice

Fengwei Zhang [1], Wei Qiao [2,3] ✉, Ji-an Wei[1,4], Zhengyi Tao [1], Congjia Chen [1], Yefeng Wu[2], Minghui Lin[1], Ka Man Carmen Ng[1], Li Zhang [4,5], Kelvin Wai-Kwok Yeung [3,6] ✉ & Billy Kwok Chong Chow [1] ✉

Secretin, though originally discovered as a gut-derived hormone, is recently found to be abundantly expressed in the ventromedial hypothalamus, from which the central neural system controls satiety, energy metabolism, and bone homeostasis. However, the functional significance of secretin in the ventromedial hypothalamus remains unclear. Here we show that the loss of ventromedial hypothalamus-derived secretin leads to osteopenia in male and female mice, which is primarily induced by diminished cAMP response element-binding protein phosphorylation and upregulation in peripheral sympathetic activity. Moreover, the ventromedial hypothalamus-secretin inhibition also contributes to hyperphagia, dysregulated lipogenesis, and impaired thermogenesis, resulting in obesity in male and female mice. Conversely, overexpression of secretin in the ventromedial hypothalamus promotes bone mass accrual in mice of both sexes. Collectively, our findings identify an unappreciated secretin signaling in the central neural system for the regulation of energy and bone metabolism, which may serve as a new target for the clinical management of obesity and osteoporosis.

As the first discovered hormone, Secretin (SCT) was originally known as a duodenal-derived humoral factor responsible for pancreatic secretion and pH neutralization in the duodenum[1]. Nevertheless, emerging data have demonstrated SCT as a pleiotropic regulator of energy homeostatic functions through the control of appetite[2–4], thermogenesis[5], and lipogenesis[6]. For example, as an anorexic peptide, SCT injection led to decreased food intake[7,8], while systemic neutralization of endogenous SCT activity increased food intake[2,5]. Moreover, postprandial elevated circulating SCT also serves as a non-adrenergic activator of thermogenesis in brown adipose tissue (BAT)[5], and intravenous infusion of SCT significantly increases the metabolic

activity of BAT and systemic energy expenditure[4]. Additionally, we further showed SCT stimulates lipolysis in white adipose tissue (WAT) via a hormone-sensitive lipase (HSL)-mediated pathway and augments intestinal lipid absorption[6]. However, despite these unveiled peripheral roles of SCT in energy metabolism, the manipulation of circulating SCT only exhibits minimal effect on body weight[5,6]. Therefore, there may exist some undiscovered mechanisms for the efficient correction of the metabolic disorder caused by the alteration in circulating SCT level.

Besides its effects on energy metabolism, many clinical studies have also suggested SCT be associated with bone homeostasis. For instance, a clinical research reported that the serum level of SCT was

[1]School of Biological Sciences, the University of Hong Kong, Hong Kong, China. [2]Applied Oral Sciences & Community Dental Care, Faculty of Dentistry, the University of Hong Kong, Hong Kong, China. [3]Shenzhen Key Laboratory for Innovative Technology in Orthopaedic Trauma, the University of Hong Kong-Shenzhen Hospital, Shenzhen, China. [4]Key Laboratory of CNS Regeneration (Ministry of Education), GHM Institute of CNS Regeneration, Jinan University, Guangzhou, China. [5]Neuroscience and Neurorehabilitation Institute, University of Health and Rehabilitation Sciences, Qingdao, China. [6]Department of Orthopaedics and Traumatology, School of Clinical Medicine, Li Ka Shing Faculty of Medicine, the University of Hong Kong, Hong Kong, China. ✉e-mail: drqiao@hku.hk; wkkyeung@hku.hk; bkcc@hku.hk

significantly lower in postmenopausal women with type I osteoporosis than those with normal bone mass[9]. In contrast, genetic over-expression of SCT contributed to alleviated type I osteoporosis and increased bone mass in mice[10]. However, supplementary administration of SCT to increase its circulating level failed to promote new bone formation[11]. These data implied that, rather than directly working on bone tissues, SCT might contribute to bone homeostasis in an indirect manner with the possible involvement of the central nervous system (CNS), which is separated from the circulation by the blood-brain barrier. In recent years, the putative neuropeptide role of SCT in CNS has drawn great attention, as it has been shown to participate in the regulation of body fluid homeostasis[12,13], motor function[14], and social behavior[15]. Using an SCT-Cre knock-in mouse model, we recently confirmed the abundance of SCTergic neurons in the ventromedial hypothalamus (VMH)[15]. As the center for homeostatic regulation in CNS[16,17], the critical roles of VMH in the control of energy metabolism and bone mass accrual have been extensively reported[18–21]. However, the functional involvement of SCT in VMH remains unclear.

In this study, we show that VMH neuron-derived SCT maintains bone and energy homeostasis by targeting SCTR-expressing cells in VMH. On the one hand, the loss of VMH-derived SCT achieved by systemic or conditional deletion significantly increased sympathetic nerve activity (SNA), leading to considerable bone loss in both male and female mice. A similar osteopenia phenotype resulting from excessive SNA is reproduced by conditional deletion of SCTR in VMH. Moreover, VMH-specific overexpression of SCT tunes down sympathetic tone and contributes to bone mass accrual. On the other hand, SCT also acts within VMH to maintain energy through the control of appetite. VMH-specific depletion of SCT or SCTR contributes to continuous hyperphagia, leading to an obese phenotype following lipogenesis dysregulation and thermogenesis impairment. Collectively, our results uncovered a previously unknown SCT signaling in VMH essential for the control of energy metabolism and bone homeostasis, which can serve as a therapeutic target for obesity or osteoporosis.

## Results

### Loss of SCT signaling leads to hyperphagia and osteopenia

To study the effect of SCT on energy metabolism, standard rodent chow-fed wild-type (WT) control mice and systemic knockout (KO) of SCT (Sct$^{-/-}$) and SCTR (Sctr$^{-/-}$) mice were housed until the age of 20 weeks. We found that the daily food intake of *ad libitum*-fed Sct$^{-/-}$ and Sctr$^{-/-}$ mice was significantly higher than that in WT mice, particularly on the third day of the test (Fig. 1a). Similarly, the rebound food intake of fasted Sct$^{-/-}$ and Sctr$^{-/-}$ mice was also significantly increased compared to WT mice (Fig. 1b, c). However, the body weight of Sct$^{-/-}$ and Sctr$^{-/-}$ mice remain intriguingly unchanged compared with WT mice throughout the 20-week period of time (Fig. 1d). Nuclear magnetic resonance (NMR) body composition analysis suggested that both Sct$^{-/-}$ and Sctr$^{-/-}$ mice even gained less fat mass fractions and more lean mass fractions compared with the WT mice (Fig. 1e). We then showed that neither Sct$^{-/-}$ nor Sctr$^{-/-}$ mice showed any abnormality in glucose (Supplementary Fig. 2a, b) and insulin tolerance (Supplementary Fig. 2c, d). Furthermore, female systemic KO mice showed similar phenotypic changes as males in appetite, body weight, and body composition (Supplementary Fig. 1a–e).

Next, we determined the effects of systemic SCT signaling on energy metabolism when Sct$^{-/-}$, Sctr$^{-/-}$, and WT mice were fed with a 60% high-fat diet (HFD). Diet-induced obesity (DIO) was prominent in all three lines of mice (Supplementary Fig. 3a). Similar to what we observed previously, systemic KO of SCT or SCTR both resulted in hyperphagia, as manifested by the significantly increased cumulative food intake by Sct$^{-/-}$ and Sctr$^{-/-}$ mice relative to the WT control (Supplementary Fig. 3b). Regardless of the overactivated appetite, the body weight of Sct$^{-/-}$ and Sctr$^{-/-}$ mice were found lower than that of WT mice starting from week 14 (Supplementary Fig. 3a). Meanwhile, after the loss of SCT, the fat mass fractions became lower while the lean mass fractions became higher

(Supplementary Fig. 3c). Then we performed indirect calorimetry studies and showed that during the dark cycle, the carbon dioxide production (VCO$_2$), oxygen consumption (VO$_2$) (Fig. 1f), and energy expenditure (EE) (Fig. 1g) of Sct$^{-/-}$ and Sctr$^{-/-}$ mice were significantly higher than those of WT mice, but their motor activity were similar (Supplementary Fig. 2e). These data, together with the increased serum level of norepinephrine (NE) (Fig. 1h), indicates an upregulation of SNA[19,22] in Sct$^{-/-}$ and Sctr$^{-/-}$ mice compared with the WT control.

As the sympathetic nervous system (SNS) is well-known to play an important role in the regulation of bone homeostasis[23–25], we then measured the bone mass at distal metaphysis of femurs using micro-computerized tomography (μCT) scans. Our data showed that systemic SCT or SCTR deficiency led to a significant decrease in the trabecular bone volume fraction (BV/TV), bone mineral density (BMD), and trabecular thickness (Tb.Th), as well as a significant increase in specific bone surface (BS/BV) (Fig. 1i, j). Moreover, the decrease in trabecular number (Tb.N), as well as the increase in trabecular pattern factor (Tb.Pf) and trabecular separation (Tb.Sp) could also be evidenced in Sctr$^{-/-}$ mice. The reduction of bone mass in Sct$^{-/-}$ and Sctr$^{-/-}$ mice was further verified by histological studies using H&E staining (Fig. 1i). Similarly, the female Sct$^{-/-}$ and Sctr$^{-/-}$ mice also exhibited reduced bone mass (Supplementary Fig. 1f, g). Taken together, we showed that SCT signaling maintains bone homeostasis through the regulation of SNA.

### VMH-derived SCT controls bone homeostasis through SNA

The VMH is well-known to be critically involved in energy and bone homeostasis through the regulation of SNA[16,17,19]. In our recent study[15], by crossing SCT-IRES-Cre knock-in mice with the R26-tdTomato reporter line, we offered a whole-brain expressional profile of SCT and identified an intensive distribution of SCTergic neurons in VMH. Here, using multiplex in situ hybridization (RNAscope), we further showed that the majority of *Sct*-positive signals co-localized with steroidogenic factor 1 (SF-1; gene name *Nr5a1*) (Fig. 2a, b and Supplementary Fig. 4a), a major marker of VMH neurons[17,19]. Immuno-fluorescent staining also confirmed the presence of SCT peptide in VMH (Fig. 2c). To investigate the physiological role of VMH-derived SCT, we elicited the virus-mediated short hairpin RNA (shRNA) approach to specifically suppress the transcription of the *Sct* gene in the VMH[26]. Bilateral injection of adeno-associated virus-ShSCT-enhanced green fluorescent protein (AAV-ShSCT-eGFP) into the VMH achieved acute VMH-SCT knockdown (KD) mice (ShSCT) (Fig. 2d, e). Meanwhile, AAV-ShCon-eGFP was used to generate sham-operated control (ShCon) mice. Here, neither TUNEL assay nor cleaved caspase-3 (an apoptosis marker[27]) immunostaining detected positive signals in the VMH of ShCon and ShSCT mice (Supplementary Fig. 5a). The expression results of NeuN, a neuronal marker[28], also showed that there was no difference in neuronal density in VMH between ShSCT and ShCon mice (Supplementary Fig. 5b, c). This indicates that ShRNA-mediated SCT ablation did not affect VMH cell viability.

Similar to systemic KO of SCT, conditional SCT KD in VMH also led to a prominent low bone mass phenotype, characterized by a significantly decreased BV/TV, BMD of TV, Tb.N, and Tb.Th, as well as a significantly increased Tb.Pf, BS/BV, and Tb.Sp at the distal metaphysis of the femur (Fig. 2f, g). Consistently, femoral bone loss in female ShSCT mice was manifested by decreased BV/TV, TV BMD, and Tb.N, and increased Tb.Pf and Tb.Sp (Supplementary Fig. 4b, c). Additionally, H&E staining not only confirmed the loss of trabecular structure but also revealed the presence of excessive lipids accumulation in bone marrow in ShSCT mice (Fig. 2h, i). Tartrate-resistant acid phosphatase (TRAP) staining showed significantly more osteoclasts in ShSCT mice than in ShCon mice (Fig. 2j). Meanwhile, the mineral apposition rate (MAR) as indicated by fluorochrome labeling was significantly lower in ShSCT mice compared with that in ShCon mice (Fig. 2k). Collectively, our results suggest that the osteopenia in ShSCT mice was caused by both increased osteoclastogenesis and decreased

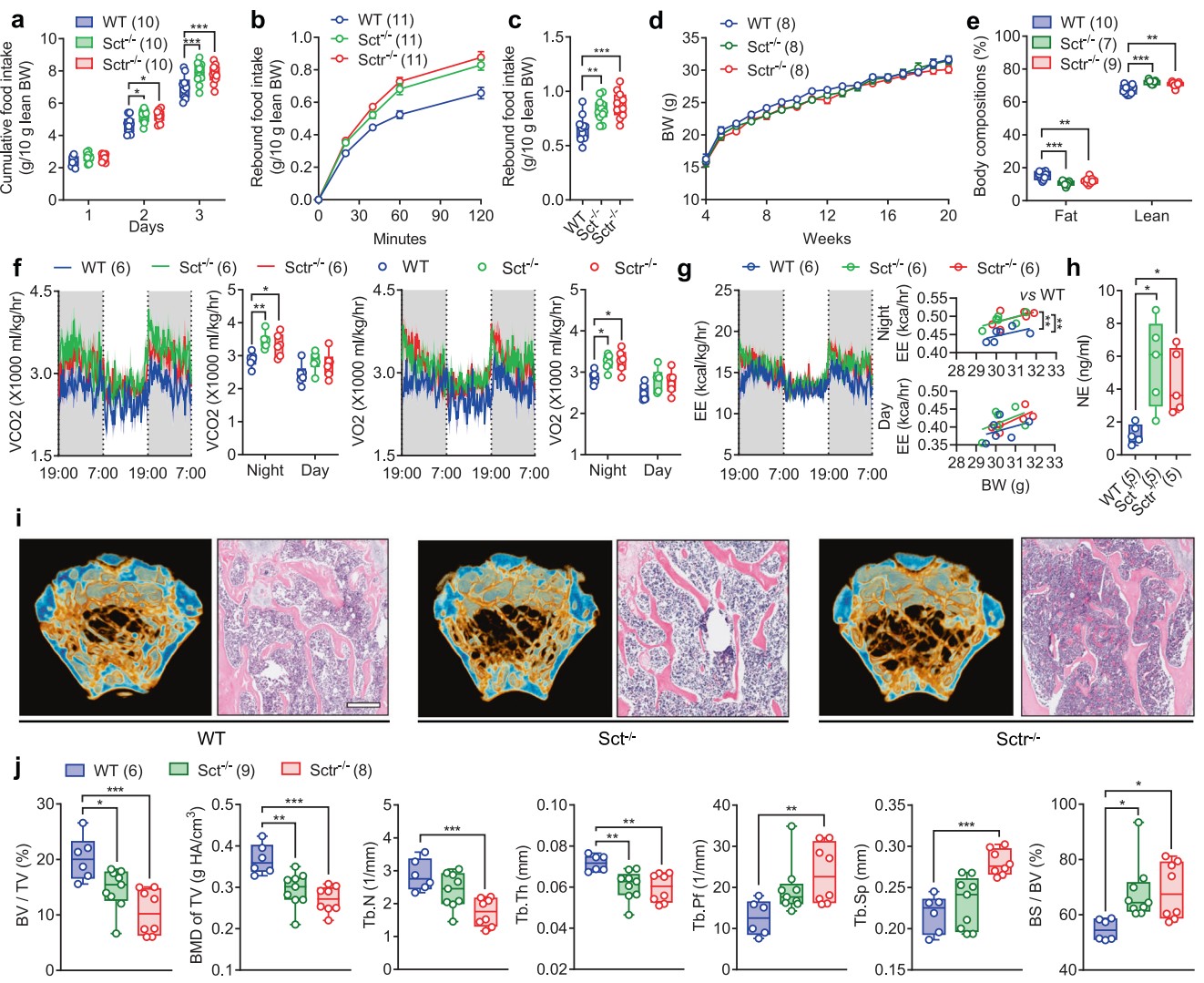

**Fig. 1 | Systemic SCT or SCTR KO results in metabolic dysfunction and bone loss. a** Daily food intake of 10-week-old WT, $Sct^{-/-}$, and $Sctr^{-/-}$ mice. Cumulative (**b**) and total (**c**) rebound food intake of 10-week-old WT, $Sct^{-/-}$, and $Sctr^{-/-}$ mice after 16 h overnight fasting. **d** Weekly body weight changes of 4- to 20-week-old WT, $Sct^{-/-}$, and $Sctr^{-/-}$ mice fed on standard rodent chow. **e** Body composition of 18-week-old WT, $Sct^{-/-}$, and $Sctr^{-/-}$ mice. **f** Temporal changes of $VO_2$ and $VCO_2$ in 16-week-old WT, $Sct^{-/-}$, and $Sctr^{-/-}$ mice. **g** Temporal changes of EE in 16-week-old WT, $Sct^{-/-}$, and $Sctr^{-/-}$ mice (night: $F_{(2, 12)}$ = 10.408, WT $vs$ $Sct^{-/-}$ (**$P_{genotype}$ = 0.0022), WT $vs$ $Sctr^{-/-}$ (**$P_{genotype}$ = 0.0028); day: $F_{(2, 12)}$ = 0.916, WT $vs$ $Sct^{-/-}$ ($P_{genotype}$ = 0.1737), WT $vs$ $Sctr^{-/-}$ ($P_{genotype}$ = 0.3896)). **h** Serum NE levels in 20-week-old WT, $Sct^{-/-}$, and $Sctr^{-/-}$ mice. **i** Representative μCT and H&E staining images showing the reduction in trabecular bone mass in the femurs of 20-week-old $Sct^{-/-}$ and $Sctr^{-/-}$ mice. Scale bar = 250 μm. **j** Corresponding measurements of (**i**): trabecular bone volume fraction (BV/TV), bone mineral density (BMD of TV), trabecular number (Tb.N), trabecular thickness (Tb.Th), trabecular pattern factor (Tb.Pf), trabecular separation (Tb.Sp), and specific bone surface (BS/BV). BW, body weight. Numbers in parentheses in each graph indicate sample size. Box plots with whiskers from minima to maxima, the central line at the 50th percentile, and the ends of the box at the 25th and 75th percentiles. **a, d, e, f** Two-way ANOVA with Holm–Šídák multiple comparisons test. **c, h, j** One-way ANOVA with Holm–Šídák multiple comparisons test. **g** One-way ANCOVA with pairwise comparisons on adjusted means. *$P$ < 0.05; **$P$ < 0.01; ***$P$ < 0.001. Error bars represent SEM. Source data are provided as a Source Data file.

new bone formation following the loss of SCT in VMH. To confirm this finding, we further generated VMH-specific SCT KD mice (SCT$^{VMH-/-}$) through bilateral injection of Cre-expressing AAV in VMH of $Sct^{fl/fl}$ mice (Fig. 2l). The mice injected with AAV-eGFP (eGFP) served as the control. We showed that the Cre recombinase efficiently mediated the deletion of the *Sct* gene in VMH (Supplementary Fig. 6a). Moreover, Cre-mediated KD of SCT in VMH reproduced the osteopenia phenotype of ShSCT mice (Fig. 2m–o). Therefore, these results suggest that SCT signaling in the VMH is responsible for the maintenance of bone mass.

Since the control of bone homeostasis by VMH is known to be mediated by the regulation of sympathetic activity following the activation of CREB signaling[20,22], we then examined the effects of SCT depletion on the phosphorylation of CREB in VMH using immunofluorescent staining. Either systemic KO of SCT or SCTR resulted in

significantly decreased phospho-CREB (pCREB) levels in VMH relative to WT mice (Supplementary Fig. 8a). Similarly, conditional KD of SCT in VMH also led to a significantly lower level of pCREB in VMH than that of the control group (Supplementary Fig. 8b), which was further confirmed by western blots (Fig. 2p). Moreover, consistent with previous findings showing CREB signaling in the VMH negatively regulates peripheral sympathetic tone[20,22,29], we found decreased pCREB levels induced by VMH-specific KD of SCT also contributed to a significantly increased serum NE level (Fig. 2q) and density of tyrosine hydroxylase (TH) positive sympathetic nerves in bone tissue (Supplementary Fig. 9) of ShSCT mice relative to their control littermates. Therefore, these results suggest that VMH-derived SCT participates in the control of bone homeostasis through the regulation of sympathetic output mediated by CREB signaling in VMH neurons.

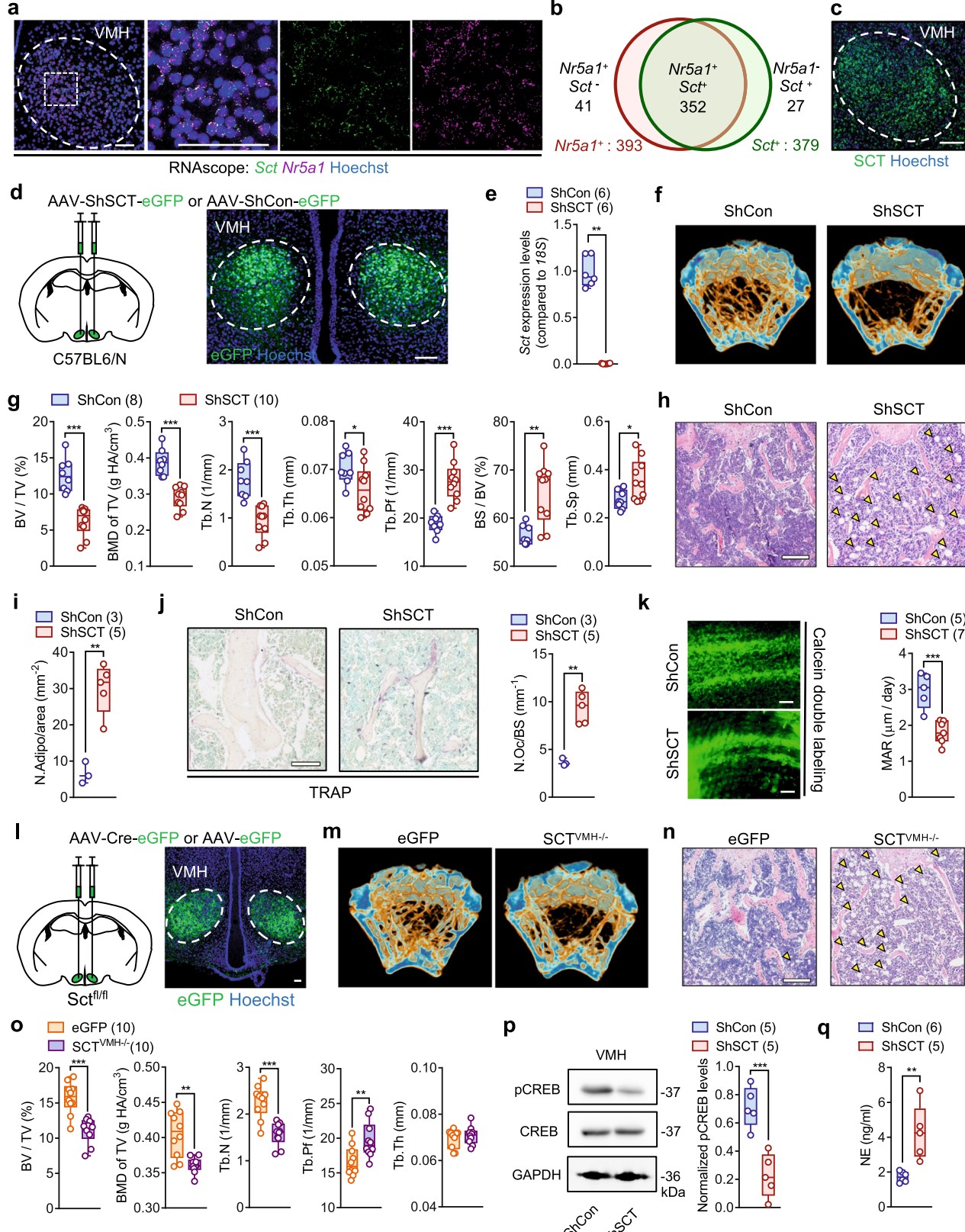

## SCT deficiency-induced hyperphagia leads to dysregulated lipogenesis

Besides its effect on bone homeostasis, we noticed that VMH-specific KD of SCT also caused a dramatic disruption of energy metabolism. Similar to the effects observed in Sct$^{-/-}$ and Sctr$^{-/-}$ mice, both male and female ShSCT mice showed increased daily food intake and rebound food intake (Fig. 3a, b and Supplementary Fig. 10a, b). Notably, although peripheral SCT administration suppressed appetite in fasted mice, it failed to completely abolish hyperphagia in ShSCT mice (Fig. 3c). Subsequent analysis of appetite-related genes revealed no

**Fig. 2 | VMH-derived SCT regulate bone mass via SNS. a** RNAscope in situ hybridization of *Sct* and *NrSa1* in the VMH sections (*n* = 3). Scale bars = 50 μm. **b** Venn diagram of the *Sct*- and *NrSa1*-positive cell distributions in (**a**). **c** Immunostaining of SCT in the VMH sections (*n* = 3). Scale bars = 100 μm. **d** Left: schematic of injecting AAV-ShSCT-eGFP/AAV-ShControl-eGFP bilaterally into the VMH of C57BL6/N mice. Right: representative image of eGFP expression in VMH (*n* = 5). Scale bars = 100 μm. **e** Reduced transcript levels of *Sct* in VMH after shRNA-mediated SCT KD. **f** Representative μCT images of femurs from 20-week-old ShSCT and ShCon littermates. **g** Corresponding measurements of (**f**). Representative femoral H&E staining (**h**) and lipid droplet statistics (**i**) of 20-week-old ShSCT and ShCon littermates. Arrows indicate lipids in bone marrow. Scale bar = 250 μm. **j** Left: TRAP staining of trabecular bone of 20-week-old ShSCT and ShCon littermates. Right: quantification of osteoclasts on the bone surface. Scale bars = 100 μm. **k** Left: calcein double labeling in trabecular bone. Right: MAR of 20-week-old ShSCT and ShCon littermates. Scale bar = 50 μm. **l** Left: schematic of injecting AAV-Cre-eGFP/AAV-eGFP bilaterally into the VMH of Sct^{fl/fl} mice. Right: representative image of eGFP expression in VMH (*n* = 5). Scale bar = 100 um. **m** Representative μCT images of femurs from 20-week-old SCT^{VMH−/−} and eGFP littermates. **n** Representative femoral H&E staining images of 20-week-old SCT^{VMH−/−} and eGFP littermates (*n* = 5). Arrows indicate lipids in bone marrow. Scale bar = 250 um. **o**, Corresponding measurements of (**m**). **p** Western blot of pCREB in VMH of 20-week-old ShSCT and ShCon littermates. **q** Serum NE levels of ShSCT and ShCon littermates. Numbers in parentheses in each graph indicate sample size. Box plots with whiskers from minima to maxima, the central line at the 50th percentile, and the ends of the box at the 25th and 75th percentiles. Two-tailed Student's *t*-test. *$P < 0.05$; **$P < 0.01$; ***$P < 0.001$. Error bars represent SEM. Source data are provided as a Source Data file.

change in the expression level of orexigenic agouti-related peptide/neuropeptide-Y (*AgRP/NPY*) but a significant decrease in the expression of anorexigenic pro-opiomelanocortin (*POMC*) in the hypothalamus of ShSCT mice (Fig. 3d), suggesting that hyperphagia may be associated with the downregulation of hypothalamic anorexia signaling. Interestingly, despite upregulated SNA, ShSCT mice displayed significantly increased body weight (Fig. 3e and Supplementary Fig. 10c), higher fat mass fractions, and lower lean mass compared with ShCon mice (Fig. 3f and Supplementary Fig. 10d). Moreover, the deletion of VMH-derived SCT contributed to a significantly increased mass of inguinal WAT (iWAT) and epididymal WAT (eWAT) (Fig. 3g), which serve as the primary WAT depots in mice[30]. In iWAT of ShSCT mice, we detected an increased number of hypertrophic adipocytes (Fig. 3h) and a significant downregulation of marker genes for thermogenic beige adipocyte[31], including uncoupling protein 1 (*Ucp1*), cytochrome c oxidase subunit 8b (*COX8b*), cell death activator CIDE-A (*CIDEA*), and fibroblast growth factor 21 (*FGF21*) (Fig. 3i), indicating the presence of metabolic dysfunction and compromised adipocyte browning process[32].

The dysregulated lipogenesis caused by the loss of VMH-derived SCT can also be evidenced in the liver, the main organ for de novo lipogenesis[33]. Macrovesicular steatosis was observed in ShSCT mice rather than ShCon mice (Supplementary Fig. 11a). Meanwhile, various hepatic lipogenesis-related genes[34,35], including peroxisome proliferator-activated receptor gamma (*PPARγ*), fatty acid synthase (*FASN*), acetyl-CoA carboxylase (*ACC*), stearoyl-CoA desaturase 1 (*SCD1*), and sterol regulatory element-binding protein 1 (*Srebp-1*) were significantly upregulated following conditional SCT KD (Supplementary Fig. 11b). Additionally, ShSCT mice also exhibited significantly raised serum insulin levels compared with ShCon (Fig. 3j). In association with hyperinsulinemia, ShSCT mice showed glucose intolerance and insulin resistance (Fig. 3k–n).

Consistent with the findings achieved by the injection of shRNA, Cre-mediated KD of SCT in VMH (SCT^{VMH−/−}) also contributed to hyperphagia (Fig. 3o, p), leading to a significant increased body weight (Fig. 3q) and fat ratios (Fig. 3r) compared to the control. Moreover, the obesity-related phenotypes observed in ShSCT mice, such as glucose intolerance and insulin insensitivity, were consistently reproduced in SCT^{VMH−/−} mice (Supplementary Fig. 6b–e). The VMH SCT deficiency induced lipogenesis in ShSCT and SCT^{VMH−/−} mice also led to a significant increase in the serum level of leptin (Supplementary Fig. 6f, g), an adipocytes-released hormone contributing to the inhibition of bone mass accrual through the upregulation of sympathetic activity[19]. Taken together, our results suggest that, VMH-derived SCT plays a critical role in the control of central perception of satiation and satiety, the loss of SCT signaling results in metabolic dysfunction and severe obesity.

## Impaired thermogenesis following SCT KD in VMH

Energy homeostasis in our body is achieved through a tightly controlled balance between energy intake and energy expenditure by the CNS[36].

The hypothalamus also elicits the SNS to regulate BAT thermogenesis to maintain the balance of energy metabolism[37,38]. However, despite the systemic upregulation of sympathetic tone, we showed the BAT thermogenesis in ShSCT and SCT^{VMH−/−} was intriguingly attenuated relative to their control littermates. VMH-specific KD of SCT led to a significantly lower $VCO_2$, $VO_2$, and EE during both dark and light cycles (Fig. 4a, b and Supplementary Fig. 6h, i). Additionally, these mice also showed significantly decreased levels of nocturnal motor activity relative to the control littermates (Fig. 4c and Supplementary Fig. 6j). Histologically, we observed an increased lipid accumulation in the interscapular BAT (iBAT) of ShSCT mice, manifested by enlarged adipocytes (Fig. 4d), increased tissue weight (Supplementary Fig. 6k), and reduced protein density (Supplementary Fig. 6l). These results consistently suggest that VMH-specific SCT KD leads to functional defects in iBAT. Indeed, compared with ShCon mice, iBAT of ShSCT mice exhibits a lower mitochondrial content (Fig. 4e) and downregulation of thermogenesis or mitochondrial function-related genes, including *Ucp1*, type II iodothyronine deiodinase (*Dio2*), peroxisome proliferator-activated receptor gamma coactivator 1α (*Pgc1α*), mitochondrial transcription factor A (*TFAM*), cytochrome c oxidase subunit 7A1 (*COX7a1*), and *COX8b* (Fig. 4f). ELISA and western blot further confirmed that the Ucp1 protein levels were reduced in ShSCT iBAT (Fig. 4g and Supplementary Fig. 6m). This is consistent with previous findings showing that genetic and/or pharmacological manipulation in the hypothalamus can lead to defected iBAT thermogenesis with increased fat mass simultaneously[39–41].

Furthermore, we showed that VMH-derived SCT deletion resulted in chronic inflammation in iBAT, as manifested by the upregulation of marker genes for macrophage (*F4/80*) and monocyte (*MCP-1* and *CD11c*) (Fig. 4h)[42,43]. Using immunofluorescent staining, we confirmed the chronic inflammation characterized by the infiltration of F4/80⁺ macrophages in iBAT of ShSCT mice (Supplementary Fig. 7a). It is thus not surprising to find that the expression of β-adrenergic receptors (*Adrb*), the primary targets respond to the regulation of the SNS[44], and tyrosine hydroxylase (*TH*), a rate-limiting enzyme in the biosynthesis of NE[45], were both significantly downregulated in iBAT of ShSCT mice compared with ShCon control (Fig. 4i, j). The impaired function of TH⁺ sympathetic nerves in iBAT following the loss of VMH-derived SCT was further confirmed by immunofluorescent staining (Supplementary Fig. 7b). Collectively, these data suggested that the deletion of SCT in VMH induced whitening and inflammation in BAT, leading to sympathetic innervation disorder and blunted response of BAT to the regulation of SNS.

## SCT signaling target SCTR-expressing cells in VMH

SCT elicits its regulatory functions following it is binding to the signaling form of its receptor, SCTR, which was previously identified in the hypothalamus using in vitro autoradiography[46], in situ hybridization[47], and RNA sequencing[48]. Here, we further confirmed the presence of SCTR in VMH by reanalyzing the published single-cell RNA sequencing dataset from a mouse model[49] (Supplementary Fig. 12) and

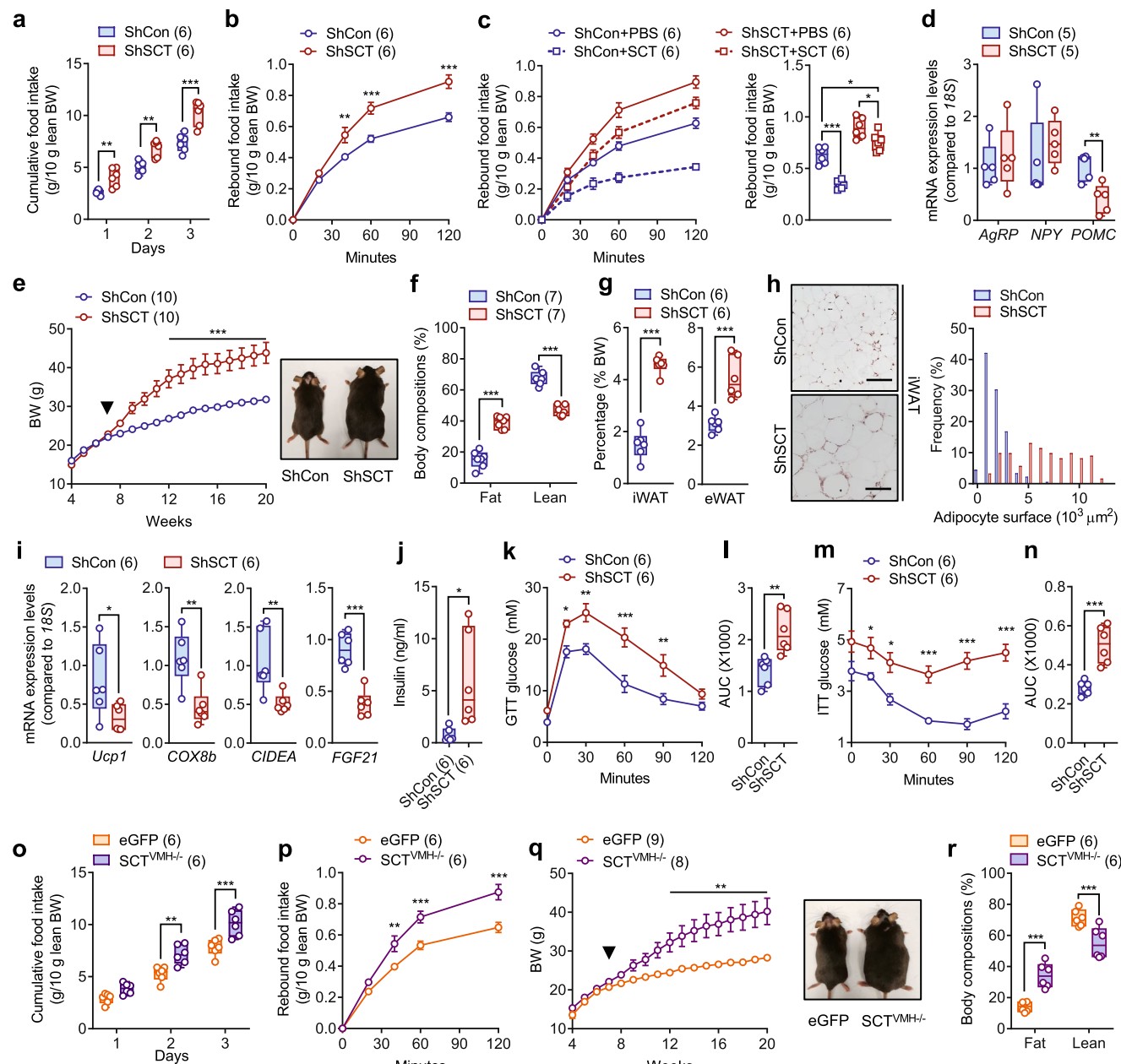

**Fig. 3 | Conditional KD of SCT in VMH leads to hyperphagia and obesity. a** Daily food intake of 10-week-old ShSCT and ShCon littermates. **b** Rebound food intake of 10-week-old overnight fasted ShSCT and ShCon littermates. **c** Cumulative (left) and total (right) rebound food intake after SCT administration in 10-week-old overnight fasted ShSCT and ShCon littermates. **d** Hypothalamic *Agrp* and *Pomc* expression of 20-week-old ShSCT and ShCon littermates. **e** Left: weekly body weight changes of ShSCT and ShCon littermates fed on standard rodent chow. Black arrow indicates virus injection at 7 weeks of age. Right: representative photographs of 20-week-old mice. **f** Body composition of ShSCT and ShCon littermates. **g** WAT mass fractions of ShSCT and ShCon littermates. **h** Left: H&E staining of iWAT from ShSCT and ShCon littermates (*n* = 5). Right: adipocyte surface area distribution of iWAT. Scale bar = 100 μm. **i** Relative expression of beige adipocyte marker genes in the iWAT. **j** Blood insulin levels of ShSCT and ShCon littermates. **k, l** Glucose tolerance test of ShSCT and ShCon littermates. **m, n** Insulin tolerance test of ShSCT and ShCon littermates. **o** Daily food intake of 10-week-old SCT^VMH−/− and eGFP littermates. **p** Rebound food intake of 10-week-old overnight fasted SCT^VMH−/− and eGFP littermates. **q** Left: weekly body weight changes of SCT^VMH−/− and eGFP littermates fed on standard rodent chow. Black arrow indicates virus injection at 7 weeks of age. Right: representative photographs of 20-week-old mice. **r** Body composition of SCT^VMH−/− and eGFP littermates. AUC, area under the curve. Numbers in parentheses in each graph indicate sample size. Box plots with whiskers from minima to maxima, the central line at the 50th percentile, and the ends of the box at the 25th and 75th percentiles. **a**, **b**, **e**, **f**, **k**, **m**, **o**–**r** Two-way ANOVA with Holm−Šídák multiple comparisons test. (**c**) One-way ANOVA with Holm−Šídák multiple comparisons test. **d**, **g**, **i**, **j**, **l**, **n** Two-tailed Student's *t*-test. \**P* < 0.05; \*\**P* < 0.01; \*\*\**P* < 0.001. Error bars represent SEM. Source data are provided as a Source Data file.

double immunofluorescent staining with glial fibrillary acidic protein (GFAP), a astrocyte marker (Fig. 5a). Both approaches indicated the expression of SCTR in astrocytes and neurons within VMH. Meanwhile, we further specified that SCT was primarily expressed in Glutamatergic neurons instead of GABAergic neurons (Supplementary Fig. 12).

To study the physiological role of SCTR in VMH, we generated VMH-specific SCTR KD mice (SCTR^VMH−/−) through bilateral injection of Cre-expressing AAV in VMH of Sctr^fl/fl mice (Fig. 5b). Meanwhile, AAV-eGFP was used to generate sham-operated control (eGFP) mice (Fig. 5b). The Cre recombinase efficiently depleted the *Sctr* gene in

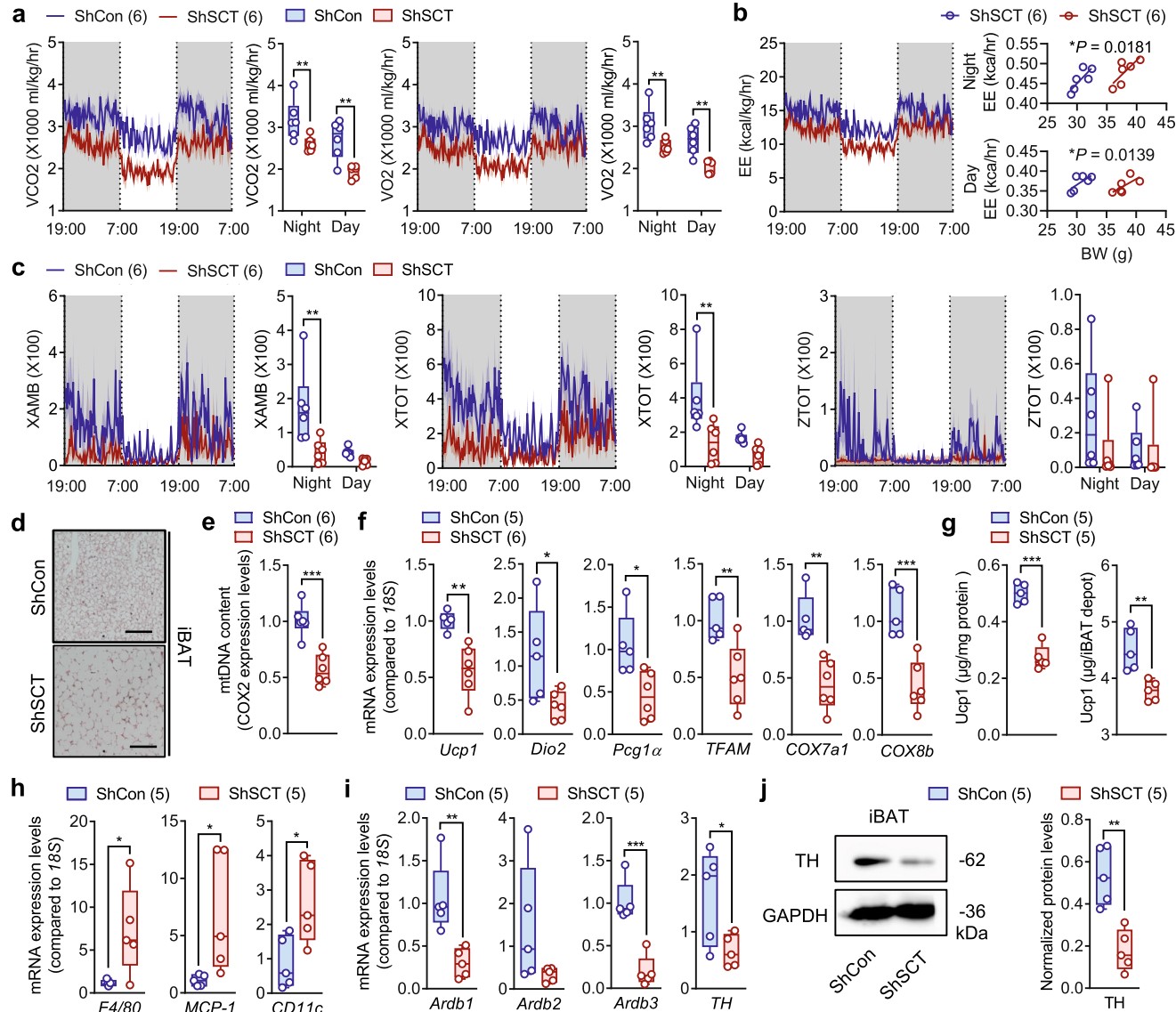

**Fig. 4 | Conditional KD of SCT in VMH leads to thermogenic dysfunction.**
**a** Temporal changes of $VO_2$ and $VCO_2$ in 16-week-old ShSCT and ShCon littermates. **b** Temporal changes of EE in 16-week-old ShSCT and ShCon littermates. **c** Temporal changes of motor activity in 16-week-old ShSCT and ShCon littermates. **d** H&E staining of iBAT. Scale bar = 100 μm. **e** Mitochondria DNA contents in iBAT. **f** Relative expression of thermogenesis-related genes in iBAT. **g** The concentration (left) and total amount (right) of Ucp1 protein in iBAT depot. **h** Relative expression of inflammatory cytokine genes in iBAT. **i** Relative expression of sympathetic innervation-related genes in iBAT. **j** Western blot of TH in iBAT. XAMB, ambulatory activity count. XTOT, total horizontal motor activity. ZTOT, total vertical motor activity. Numbers in parentheses in each graph indicate sample size. Box plots with whiskers from minima to maxima, the central line at the 50th percentile, and the ends of the box at the 25th and 75th percentiles. **a**, **c** Two-way ANOVA with Holm–Šídák multiple comparisons test. **b** One-way ANCOVA with pairwise comparisons on adjusted means. **e**–**j** Two-tailed Student's *t*-test. *$P < 0.05$; **$P < 0.01$; ***$P < 0.001$. Error bars represent SEM. Source data are provided as a Source Data file.

VMH (Fig. 5c). Cre-mediated KD of SCTR in VMH reproduced the osteopenia phenotype seen in ShSCT and SCT$^{VMH-/-}$ mice. Our μCT analysis showed that conditional KD of SCTR in VMH resulted in a considerably lower trabecular bone density in the distal metaphysis of the femur at 20 weeks of age (Fig. 5d). Quantitative data showed a significantly decreased BV/TV, BMD, Tb.N, and Tb.Th, as well as an increased Tb.Pf in the femurs of SCTR$^{VMH-/-}$ mice compared with those in the control group (Fig. 5e). Moreover, the marked reduction in bone trabecula and the diminished new bone formation following the loss of SCTR in VMH were confirmed by H&E staining (Fig. 5f) and fluorochrome labeling (Fig. 5g), respectively. Additionally, we showed the bone loss in SCTR$^{VMH-/-}$ mice was accompanied by a significant increase in the serum level of NE (Fig. 5h), suggesting that VMH-derived SCT may target the locally expressed SCTR in an autocrine or paracrine

manner to participate in the control of bone homeostasis through regulating sympathetic outflow.

Besides the effects on bone homeostasis, VMH-specific deletion of SCTR also led to a similar obese phenotype observed in ShSCT and SCT$^{VMH-/-}$ mice. Compared with eGFP mice, both daily and rebound food intake were found markedly increased in SCTR$^{VMH-/-}$ mice (Fig. 5i, j). Consequently, SCTR$^{VMH-/-}$ mice gained body weight significantly faster (Fig. 5k), with more fat mass and lower lean mass than the control mice (Fig. 5l). Concomitantly, SCTR$^{VMH-/-}$ mice exhibited not only hyperleptinemia (Supplementary Fig. 13a) but also glucose intolerance and insulin resistance (Supplementary Fig. 13b–e), as seen in ShSCT and SCT$^{VMH-/-}$ mice. Indirect calorimetry further showed that VMH-specific SCTR deletion contributed to significantly reduced $VCO_2$, $VO_2$, and EE during both dark and light cycles

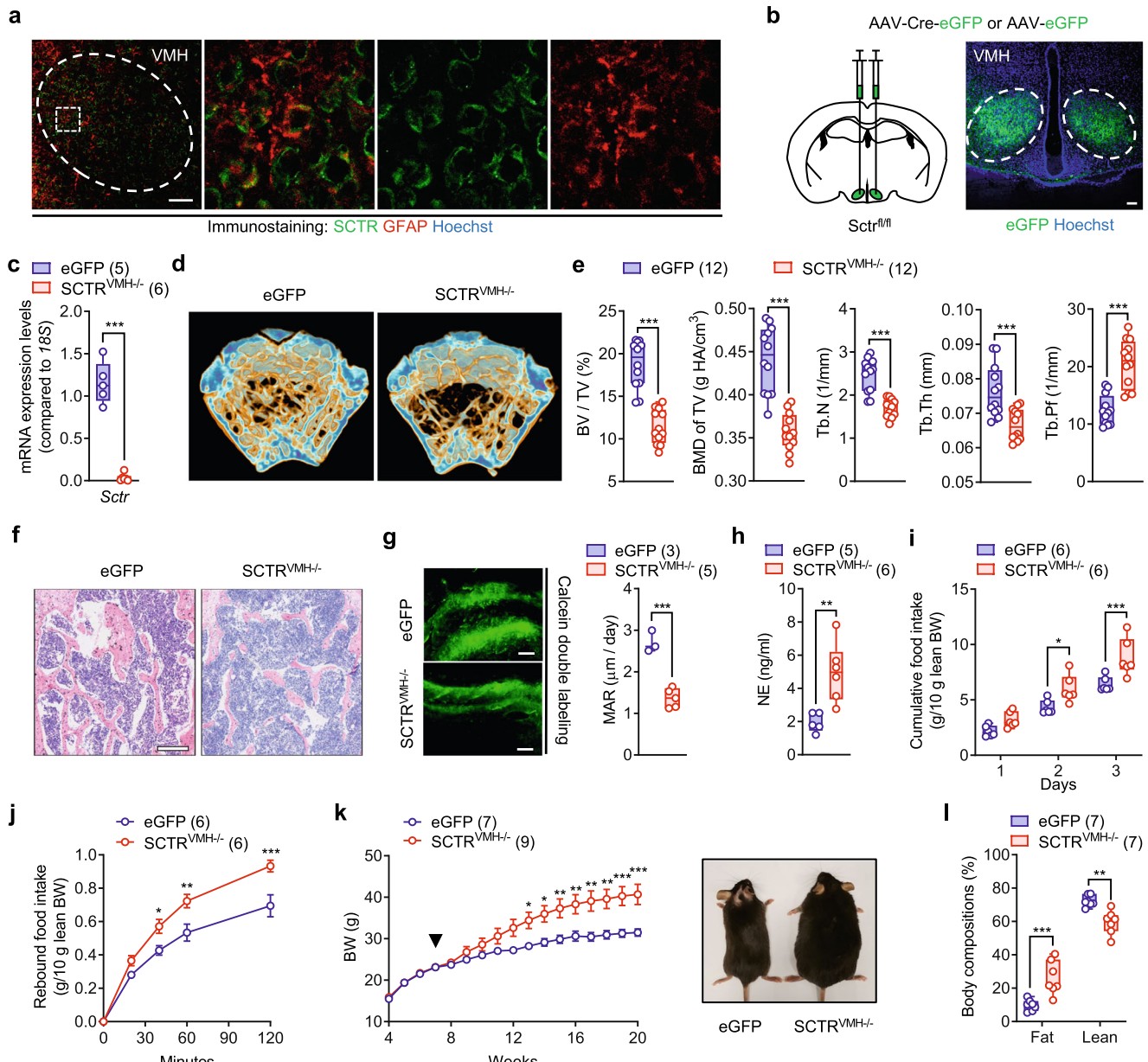

**Fig. 5 | Conditional KD of SCTR in VMH leads to osteopenia, hyperphagia, and obesity. a** Double-immunostaining of SCTR and GFAP in the VMH sections (n = 5). Scale bar = 100 μm. **b** Left: schematic of injecting AAV-Cre- eGFP/AAV-eGFP bilaterally into the VMH of Sctr[fl/fl] mice. Right: representative image of eGFP expression in VMH (n = 5). Scale bar = 100 μm. **c** Reduced transcript levels of *Sctr* in VMH after Cre-mediated SCTR KD. **d** Representative μCT images of femurs from 20-week-old SCTR[VMH-/-] and eGFP littermates. **e** Corresponding measurements of (**d**). **f** Representative femoral H&E staining images of 20-week-old SCTR[VMH-/-] and eGFP littermates. Scale bar = 250 μm. **g** Left: calcein double labeling in trabecular bone. Right: MAR of 20-week-old SCTR[VMH-/-] and eGFP littermates. Scale bar = 50 μm. **h** Serum NE levels of SCTR[VMH-/-] and eGFP littermates. **i** Daily food intake of 10-week-old SCTR[VMH-/-] and eGFP littermates. **j** Rebound food intake of 10-week-old overnight fasted SCTR[VMH-/-] and eGFP littermates. **k** Left: weekly body weight changes of SCTR[VMH-/-] and eGFP littermates fed on standard rodent chow. Black arrow indicates virus injection at 7 weeks of age. Right: representative photographs of 20-week-old mice. **l** Body composition of 18-week-old SCTR[VMH-/-] and eGFP littermates. Numbers in parentheses in each graph indicate sample size. Box plots with whiskers from minima to maxima, the central line at the 50th percentile, and the ends of the box at the 25th and 75th percentiles. **c**, **e**, **g**, **h** Two-tailed Student's *t*-test. **i**–**l** Two-way ANOVA with Holm–Šídák multiple comparisons test. *P < 0.05; **P < 0.01; ***P < 0.001. Error bars represent SEM. Source data are provided as a Source Data file.

(Supplementary Fig. 13f, g), as well as blunted nocturnal activity levels (Supplementary Fig. 13h). Together, we showed the VMH-derived SCT directly acts on SCTR located within VMH to regulate the bone and energy homeostasis.

### VMH-specific deletion of SCT signaling exacerbates obesity and osteopenia in DIO mice

Previous studies in animal models and human participants have demonstrated that excess visceral fat may contribute to bone loss,

indeed, obesity is closely related to bone homeostasis through various signaling pathways involved in metabolic disorders[50–53]. Therefore, we sought to investigate whether VMH-specific SCT KD further exacerbates the metabolic disorders and bone loss induced by DIO. Both SCT[VMH-/-] and control littermates gained weight rapidly after being fed with HFD, however, SCT[VMH-/-] mice had a larger body weight than eGFP mice starting from the third week after HFD feeding (Fig. 6a). Nonetheless, the differences in body composition (Fig. 6b) and serum leptin levels (Fig. 6c) between SCT[VMH-/-] and eGFP littermates were not

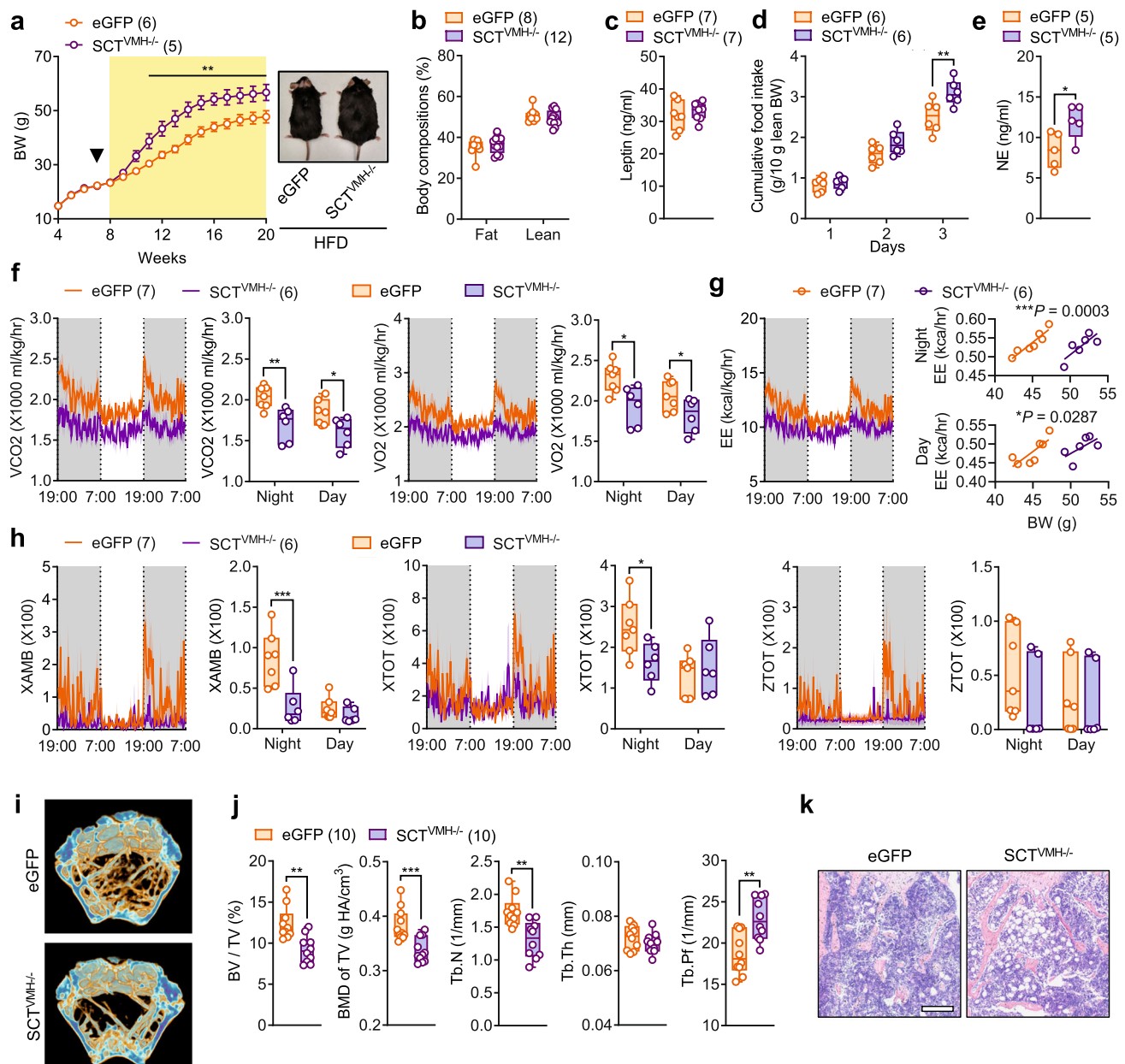

**Fig. 6 | Conditional KD of SCT in the VMH exacerbates obesity and osteopenia in DIO mice. a** Left: weekly body weight changes of SCT^VMH−/− and eGFP littermates fed on HFD. Black arrow indicates virus injection at 7 weeks of age. The yellow shaded area indicates that the mice were fed HFD. Right: representative photographs of 20-week-old mice. **b** Body composition of 18-week-old HFD-fed SCT^VMH−/− and eGFP littermates. **c** Serum leptin levels in HFD-fed SCT^VMH−/− and eGFP littermates. **d** Daily food intake of 16-week-old HFD-fed SCT^VMH−/− and eGFP littermates. **e** Serum NE levels in HFD-fed SCT^VMH−/− and eGFP littermates. **f** Temporal changes of VCO₂ and VO₂ in 16-week-old HFD-fed SCT^VMH−/− and eGFP littermates. **g** Temporal changes of EE in 16-week-old HFD-fed SCT^VMH−/− and eGFP littermates. **h** Temporal changes of motor activity in 16-week-old HFD-fed SCT^VMH−/− and eGFP littermates.

**i** Representative μCT images of femurs from 20-week-old HFD-fed SCT^VMH−/− and eGFP littermates. **j** Corresponding measurements of (**i**). **k** Representative femoral H&E staining images of 20-week-old HFD-fed SCT^VMH−/− and eGFP littermates. Scale bar = 250 μm. Numbers in parentheses in each graph indicate sample size. Box plots with whiskers from minima to maxima, the central line at the 50th percentile, and the ends of the box at the 25th and 75th percentiles. **a**, **b**, **d**, **f**, **h** Two-way ANOVA with Holm–Šídák multiple comparisons test. **c**, **e**, **j** Two-tailed Student's *t*-test. **g** One-way ANCOVA with pairwise comparisons on adjusted means. *$P < 0.05$; **$P < 0.01$; ***$P < 0.001$. Error bars represent SEM. Source data are provided as a Source Data file.

significant. Similar to standard rodent chow-fed mice, HFD-fed SCT^VMH−/− mice also exhibited significantly increased daily food intake (Fig. 6d) and higher sympathetic tone compared with the eGFP control (Fig. 6e). Meanwhile, indirect calorimetry revealed that SCT^VMH−/− mice showed significantly decreased VCO₂, VO₂, and EE (Fig. 6f, g) as well as nocturnal activity levels (Fig. 6h) relative to eGFP mice. Moreover, when challenged by HFD, the Cre-mediated KD of SCT in VMH contributed to

further bone loss accompanied by more severe lipid infiltration compared with their control littermate (Fig. 6i–k).

Similarly, VMH-specific KD of SCTR in DIO mice also led to significantly more serious osteopenia (Supplementary Fig. 14a–c) accompanied by the upregulation of sympathetic tone (Supplementary Fig. 14d). Meanwhile, HFD-fed SCTR^VMH−/− mice also exhibited significantly increased cumulative food intake (Supplementary

Fig. 14e) and body weight (Supplementary Fig. 14f) compared with their control littermates. We also observed impaired EE (Supplementary Fig. 14g), glucose intolerance (Supplementary Fig. 14h, i), and insulin insensitivity (Supplementary Fig. 14j, k) in HFD-fed SCTR[VMH−/−] mice. Thus, we showed that the loss of SCT signaling in VMH under obesity situation can be a major concern as it deteriorates the dysfunctions in energy and bone metabolism.

### SCT overexpression in VMH promotes bone mass accrual

Since SCT insufficiency in VMH has been shown to induce bone loss, we asked whether the supplement of SCT peptide in VMH can contribute to bone mass accrual. To address this question, we bilaterally injected an rAAV carrying an elongation factor 1 (EF1α) promoter element into the VMH of 7-week-old C57BL6/N mice to achieve site-specific overexpression of SCT (SCToe) (Fig. 7a). Mice injected with rAAV-EF1α-mCherry (mCherry) serve as the control. Mice were fed standard rodent chow until 20 weeks of age. Following the increase of SCT level in VMH (Fig. 7b), the pCREB levels increased in VMH of the SCToe mice (Fig. 7c). Correspondingly, the sympathetic tone in SCToe mice was significantly attenuated, as evidenced by the reduced serum NE levels relative to the mCherry control (Fig. 7d). Consequently, the bone density in the distal metaphysis of the femur determined by μCT was found significantly higher in SCToe mice compared with that in control mice (Fig. 7e, f). The increment in bone trabeculae structures is further confirmed by the histological study (Fig. 7g). Complementary to this, the number of TRAP-positive cells was found significantly decreased in SCToe mice relative to mCherry mice (Fig. 7h), suggesting attenuated osteoclastogenesis activities. Consistently, the female SCToe mice also exhibited increased bone mass (Supplementary Fig. 15). In addition to its effects on bone homeostasis, VMH-specific overexpression of SCT seemed to have less influence on energy balance, as the appetite (Fig. 7i, j), body weight (Fig. 7k), and body composition (Fig. 7l) remained unchanged after the increase of SCT in VMH. This implies that, unlike the SCT-driven central control of bone homeostasis tightly mediated by SNS, the VMH perception of satiation can be achieved with a minimal level of SCT, thus additional SCT exceeding this level barely contributes to the regulation of appetite and energy metabolism. Taken together, these results suggest that SCT overexpression in VMH reduces bone resorption and promotes bone mass accrual in mouse femurs.

## Discussion

The anorexigenic effect of SCT has been extensively reported, as peripheral or intracerebroventricular SCT administration effectively reduced food intake in fasted sheep[8], rats[54], and mice[2,3]. A recent study proposed that postprandial upregulation of gut-derived SCT acts on BAT to initiate thermogenesis and satiation through an unclearly defined afferent feedback to the brain[5]. Their follow-up study in human participants also showed SCT reduces central responses to appetizing food and delays the motivation to refeed after a meal[4]. However, the mechanism through which these peripheral stimulations signal the CNS to induce satiation remains unanswered. In this study, we showed VMH[SCT] neurons overlap with VMH[SF-1] neurons, which are generally known to control appetite[55,56]. Distinct from the canonical roles in circulation as a hormone, SCT in VMH serves as a neuropeptide to regulate appetite in a top-down manner. This central anorexia signal is critical to energy metabolism because the loss of SCT signaling in VMH quickly leads to hyperphagia and severe obesity, which was never seen before through the manipulation of peripheral SCT level[5,6]. Using our VMH-specific KD model, we also showed the hyperphagia and obesity induced by the loss of VMH-derived SCT can barely rescued by peripherally produced or exogenous SCT. Moreover, our data also suggests that SCT signaling-driven satiation is mediated by the anorexigenic POMC pathway rather than the orexigenic AgRP/NPY

pathway. This is consistent with previous neural mapping findings showing VMH neurons inhibit feeding by sending strong excitatory inputs to POMC neurons in the arcuate nucleus (ARC), whereas ARC[NPY] neurons do not receive innervations from the VMH[57]. Indeed, specific gene deletion in VMH attenuated glutamatergic input[58] or POMC mRNA expression[18] in ARC[POMC] neurons. Here, we show that VMH-specific SCT KD reduces local CREB phosphorylation levels, which is known to positively correlate with neuronal activity[59] and glutamate transporter levels[60]. Thus, VMH[SCT] neurons may innervate the activity of ARC[POMC] neurons by mediating glutamatergic signaling, which requires further investigation.

The VMH is designated as a hub for sensing and integrating metabolic signals, as well as regulating peripheral organs to maintain our body weight[17,61,62]. Previous studies have reported that electrolytic lesions of VMH[63] or cell-specific gene deletion in VMH[17,18,45] could induce obesity. Here, we unveiled the critical role of VMH SCT signaling in energy metabolism using multiple animal models, including Cre-loxP recombination-mediated conditional SCT/SCTR KD and RNA interference-induced SCT KD. They consistently led to severe obese phenotype following dysregulated lipogenesis and impaired thermogenesis. Nevertheless, standard rodent chow-fed systemic KO mice had normal body weight but reduced fat content and increased food intake. Although nothing similar has been reported yet, one study showed that neutralizing endogenous SCT can acutely increase appetite, suggesting that blocking SCT signaling globally can alter eating behavior. The difference in the metabolic phenotypes induced by VMH-specific or systemic deletion of SCT suggests that the central SCT derived from VMH and the circulating SCT, which is primarily secreted by the digestive system, may elicit different pathways to regulate energy balance collaboratively. Indeed, over the decades, SCT was better known as a duodenum-derived exocrine hormone responsible for digestion and absorption of macronutrients, including fat and protein[64–66]. In humans, SCTR expression in omental fat is positively correlated with body mass index[67]. The essential roles of SCT in fatty acid/glucose uptake and adipogenesis have also been extensively studied recently[6,67]. It is thus not surprising that Sct[−/−] and Sctr[−/−] mice were resistant to body fat gain or obesity progression due to impaired intestinal lipid absorption, as we reported previously[6].

Although it is generally accepted that adipose tissue is innervated by the sympathetic nerves and the upregulation of SNA initiates energy expenditure through thermogenesis and lipolysis[45,68]. However, in our study, several different mouse models exhibited significantly lower energy expenditure following VMH-specific SCT signaling deletion, regardless of the increase in SNA. Further investigation showed the BAT presented a chronic inflammation status characterized by the infiltration of adipose tissue macrophage, which might be metabolically activated by the elevated glucose and insulin resulting from hyperphagia[69]. Gene expression analysis confirmed the presence of macrophages and other immune cell infiltration in iBAT of ShSCT mice, which may cause whitening and even death of brown adipocytes[42]. Therefore, the compromised BAT thermogenic function can be caused by low-grade adipose tissue inflammation, as described previously[42]. Moreover, we also detected a decreased expression of β-adrenoceptors and TH in the iBAT of ShSCT mice, indicating a blunted response and innervation of SNS. This can be explained by the desensitization of BAT induced by chronic sympathetic overactivity, which was seen in an anti-obesity clinical trial indicating the long-term use of sympathomimetic drug ephedrine can lead to decreased basal BAT activity and reduced thermogenic responses to adrenergic stimuli[70].

In addition to its effects on energy metabolism, the SNS has also been known as a regulator of bone homeostasis, because the NE released by sympathetic neurons inhibits osteogenesis and accelerates osteoclastic activity[23,71]. Although some earlier studies found that SCT

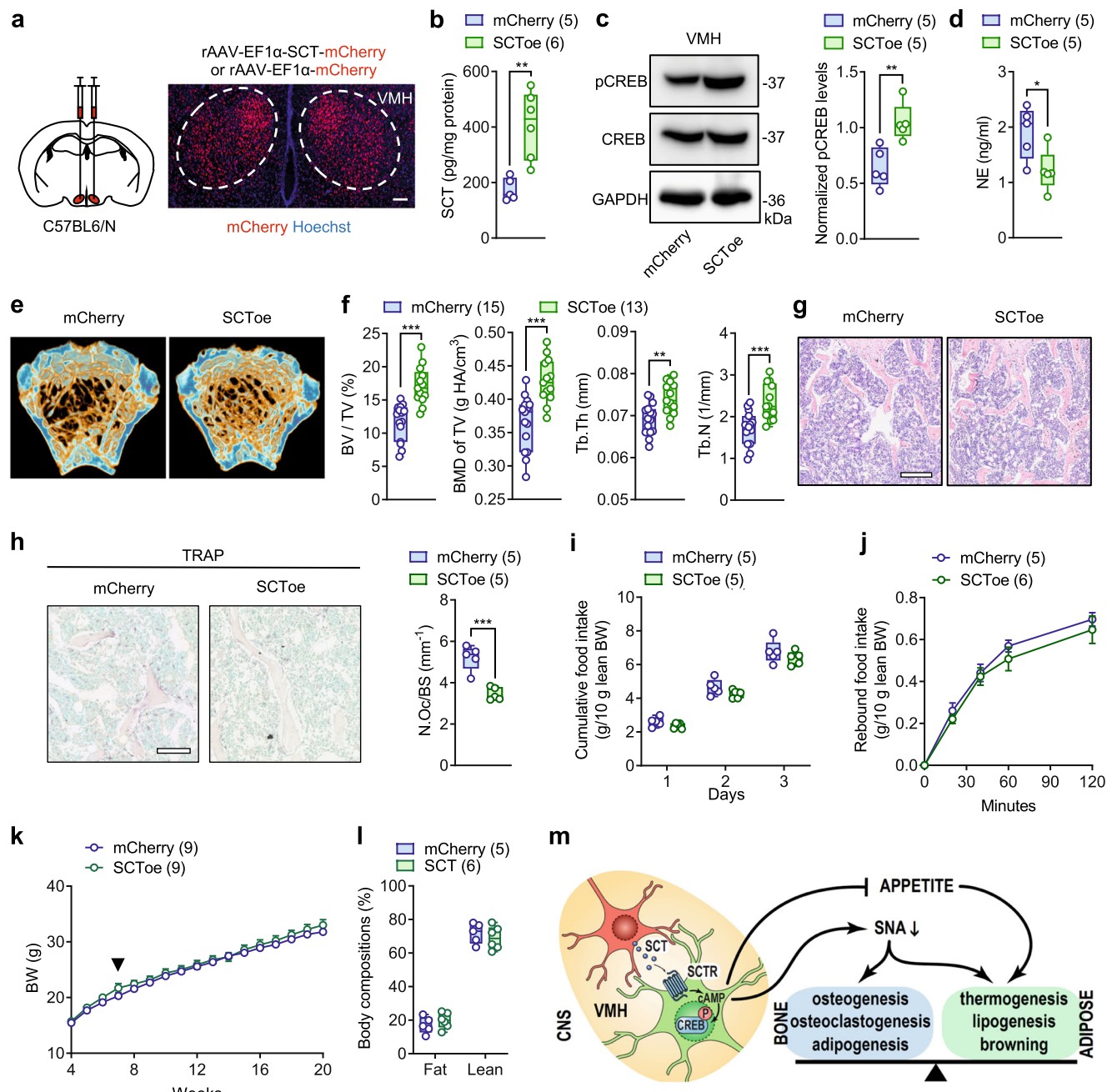

**Fig. 7 | SCT overexpression in VMH leads to bone mass increment. a** Left: schematic of injecting rAAV-EF1α-SCT-mCherry or rAAV-EF1α-mCherry bilaterally into the VMH of C57BL6/N mice. Right: representative image of mCherry expression in VMH (n = 5). Scale bars = 100 μm. **b** Enhanced SCT levels in VMH after EF1α-mediated SCT overexpression. **c** Serum NE levels in SCToe and mCherry littermates. **d** Western blot of pCREB in VMH of 20-week-old SCToe and mCherry littermates. **e** Representative μCT images of femurs from 20-week-old SCToe and mCherry littermates. **f** Corresponding measurements of (**e**). **g** Representative femoral H&E staining images of 20-week-old SCToe and mCherry littermates. Scale bar = 250 μm. **h** Left: TRAP staining of trabecular bone of 20-week-old SCToe and mCherry littermates. Right: quantification of osteoclasts on the bone surface. Scale bars = 100 μm. **i** Daily food intake of 10-week-old SCToe and mCherry littermates.

**j** Rebound food intake of 10-week-old overnight fasted SCToe and mCherry littermates. **k** Weekly body weight changes of SCToe and mCherry littermates fed on standard rodent chow. Black arrow indicates virus injection at 7 weeks of age. **l** Body composition of 18-week-old SCToe and mCherry littermates. **m** The schematic shows the mechanism by which VMH-derived SCT signaling regulates skeletal and metabolic homeostasis. Numbers in parentheses in each graph indicate sample size. Box plots with whiskers from minima to maxima, the central line at the 50th percentile, and the ends of the box at the 25th and 75th percentiles. (**b**, **c**, **d**, **h**) Two-tailed Student's *t*-test. (**i**, **j**, **k**, **l**) Two-way ANOVA with Holm–Šídák multiple comparisons test. *$P < 0.05$; **$P < 0.01$; ***$P < 0.001$. Error bars represent SEM. Source data are provided as a Source Data file.

could mediate catecholamine synthesis in sympathetic nerve endings by activating TH[72], chronic subcutaneous SCT administration in a recent study failed to alter the bone phenotype[11]. Moreover, considering the absence of SCTR in bone cell lineage[11], it is unlikely that SCT can directly regulate bone cells in the skeletal microenvironment.

In this study, we showed that the loss of SCT/SCTR in VMH induced by systemic KO or VMH-specific KD consistently led to osteopenia following the upregulation of sympathetic tone. In contrast, SCT overexpression in VMH tunes down sympathetic tone to promote bone mass accrual. Moreover, our data suggests the lack of central SCT can

barely be compensated by the presence of peripheral SCT. The discovery of the central role of SCT in the regulation of bone homeostasis further supports recent insights on the brain-bone axis, highlighting the tight connection between CNS and the skeletal system[16,73]. Another interesting finding arises from this study is that the SCT-deficiency induced metabolic dysfunction also led to higher plasma insulin/leptin levels associated with obesity, which may further contribute to the upregulation of sympathetic tone and deteriorated osteopenia[20]. In fact, HFD models with similar leptin levels phenocopied the skeletal changes shown in conventional diet models, suggesting that loss of SCT signal in the VMH should be a major contributor to bone loss even in DIO mice. Additionally, it's noteworthy that an increased bone marrow adiposity was observed in VMH-specific SCT KD mice. This can be caused by metabolism disorder as insulin resistance associated with obesity can contribute to bone marrow adiposity[74]. Meanwhile, the upregulation of SNA can also promote adipogenesis in bone marrow because the lineage commitment of mesenchymal stromal cells could be shifted by the alteration in sympathetic tone[75].

In summary, this study reveals the neural mechanisms by which central SCT participates in homeostatic regulation. Specifically, VMH-derived SCT controls food intake and prevents hyperphagia by activating the hypothalamic POMC pathway and regulates SNA by promoting CREB phosphorylation, both of which are jointly involved in maintaining the balance between osteogenesis, osteoclasts, thermogenesis, and lipogenesis. Thus, our findings identify a required SCT signaling in VMH for the regulation of energy and bone metabolism, which may serve as a new target for the clinical management of obesity and osteoporosis (Fig. 7m).

## Methods

### Ethics statement
Animal care, welfare monitoring, experimental procedures, and euthanasia practices were carried out with the protocols approved by the Committee on the Use of Live Animals in Teaching and Research (CULATR) of the University of Hong Kong (protocol No. 5791-21). All animals were maintained in a facility accredited by the Association for the Assessment and Accreditation of Laboratory Animal Care International (AAALAC). This study follows ARRIVE guidelines.

### Animals
All mice used in this study were maintained in the C57BL6/N genetic background. Sct$^{-/-}$[76], Sctr$^{-/-}$[77], Sct$^{fl/fl}$[76], and Sctr$^{fl/fl}$[12] mouse strains were previously described. All experiments were carried out using ≥4-week-old male and female mice. Unless otherwise specified, the mice referred to in the study are male. All mice were housed in temperature-controlled (20–26 °C) and humidity-controlled (30–70%) rooms with a 12:12 h light: dark cycle with *ad libitum* access to standard rodent chow (0.3% Na; 5010, LabDiet) or HFD (60% fat, D12492, Research Diets) and water unless otherwise specified. In this study, mice were euthanized with $CO_2$.

### Viral constructs
The following viruses were purchased from BrainVTA: rAAV2/9-hSyn-CRE-EGFP (PT-1168), $5 \times 10^{12}$ viral genomes per ml. rAAV2/9-hSyn -EGFP (PT-1990), $5 \times 10^{12}$ viral genomes per ml. rAAV2/9-EF1a-secretin-P2A-mCherry (PT-2581), $5 \times 10^{12}$ viral genomes per ml. rAAV2/9-EF1a-P2A-mCherry (PT-1940), $5 \times 10^{12}$ viral genomes per ml. The following AAV virus was purchased from BrainCase: rAAV-U6-shRNA(mSecretin)-CMV-EGFP (BC-0566), $5 \times 10^{12}$ viral genomes per ml. rAAV-U6-shRNA(Scramble)-CMV-EGFP (BC-0186), $5 \times 10^{12}$ viral genomes per ml.

### Surgery
According to the purpose of different experiments, 7-week-old WT, Sct$^{fl/fl}$, or Sctr$^{fl/fl}$ mice were used for stereotaxic surgery. Mice were anaesthetized by intraperitoneal (i.p.) injection (10 µl/g BW) of a mixture of ketamine (1 mg/ml) and xylazine (10 mg/ml). The mice were then placed in a stereotaxic apparatus (RWD Life Science, 68513). An incision was made to expose the skull. A small craniotomy, less than 1 mm, was made using a hand drill at the regions of interest. For microinjections, a Hamilton syringe (65458-02) filled with AAV virus or tracer was placed into the target area according to the corresponding coordinates: VMH (−1.58 mm antero-posterior (AP), ±0.4 mm medio-lateral (ML), −5.65 mm dorso-ventral (DV), relative to bregma). 150 nl virus were injected at speed of 50 nl/min. The needle was left in place for an additional 10 min before retraction. The scalp incision was sutured, and postinjection analgesics were given for 3 days to aid recovery. All mice were placed in an incubator (33 °C) overnight to recover, and were then housed in the animal facility.

### Body weight and compositions
The body weights of mice were monitored weekly from weaning (4 weeks old). For the HFD study, mice were maintained on the standard rodent chow diet until 8-week-old, then were switched to HFD for an additional 12 weeks until 20-week-old. Body compositions of all mice were analyzed by NMR (LF90 Minispec, Bruker Corp.) at 18 weeks of age.

### Metabolic cage study
For metabolic cage studies, 16-week-old male mice were used. Metabolic rates were assessed with an indirect calorimetry system (Comprehensive Lab Animal Monitoring System: Oxymax®-CLAMS, Columbus Instruments). For environmental acclimation, first, the experimental mice were housed 4 hours in metabolic cages individually and maintained in the same room where the metabolic analyses were performed. Then the mice were individually housed in the metabolic chambers and acclimated for 4 h. After acclimation in the chamber, oxygen consumption ($VO_2$), carbon dioxide production ($VCO_2$), heat generation (EE), and movement were measured by Oxymax (v 5.0). Diet and water were available *ad libitum* unless otherwise indicated.

### Glucose and insulin tolerance tests
For glucose tolerance test (GTT), 16-week-old mice were fasted overnight for 16 h and provided with water *ad libitum*. The next day, mice were housed in individual cages and allowed to acclimate for 2 h followed by i.p. injection of 1.0 g/kg glucose (G8270, Millipore). For insulin tolerance test (ITT), 16-week-old mice were fasted overnight for 16 hours in individual cages with free access to water. Insulin (0.8 U/kg, I0908, Sigma-Aldrich) was administered by i.p. injection. Blood samples were obtained from a tail nick, and blood glucose was measured at 0, 15, 30, 60, 90, and 120 minutes using a commercial glucometer (Accu-Chek® Guide, Roche).

### ELISA assays
Blood was collected from 20-week-old male mice. Serum NE (KA1891, Abnova), SCT (EK-067-04, Phoenix Pharmaceuticals), insulin (90080, Crystal Chem), and leptin (90030, Crystal Chem) were analyzed using the corresponding ELISA kits following the manufacturer's instructions. The Ucp1 concentration was measured using an ELISA kit (SEF557Ra, Cloud-Clone Corp.) following the manual instruction. The total Ucp1 content of iBAT depot was calculated as iBAT depot mass (mg) * iBAT protein density (µg/mg iBAT) * Ucp1 concentration (µg/µg iBAT protein). Total iBAT proteins were extracted using RIPA lysis buffer (89900, ThermoFisher). Protein density was measured using the BCA Protein Assay Kit (P0011, Beyotime).

### Immunohistochemistry
For brain immunofluorescence study, mice were deeply anaesthetized with isoflurane and then transcardially perfused with phosphate buffered saline (PBS) followed by 4% paraformaldehyde in PBS (PFA, pH

7.4) at 4 °C. The brains were extracted and fixed in PFA overnight at 4 °C and then cryo-protected in 30% sucrose overnight at 4 °C. Free-floating sections (35 μm) were prepared with a cryostat (CM3050S, Leica) for antibody staining. Other tissues including bone, iWAT, iBAT, and liver were dissected and post-fixed in 4% PFA overnight at 4 °C. Tissue samples were then paraffin embedded and cut into 5 μm slices. After blocking with a blocking buffer containing 5% bovine serum albumin (10738328103, Roche) and 0.3% Triton X-100 in PBS (PBST) at room temperature for 1 h, brain sections were incubated with primary antibodies in blocking buffer at 4 °C for 12 h. After washing three times with PBS, the sections were incubated with secondary antibodies in blocking buffer for 4 h. The primary antibodies (1:500 dilution) used in this study were as follows: rabbit anti-SCTR (HPA007269, Sigma-Aldrich), rabbit anti-SCT (G-067-04, Phoenix Pharmaceuticals), rabbit anti-NeuN (ABN78, Millipore), rabbit anti-Cleaved Caspase-3 (Asp175) (9661, Cell Signalling Technology), rabbit anti-pCREB (phospho S133) (ab32096, Abcam), mouse anti-CREB (ab178322, Abcam), rabbit anti-TH (AB152, Millipore), rabbit anti-F4/80 (70076, Cell Signalling Technology), and chicken anti-GFAP (ab4674, Abcam). The secondary antibodies (1:500 dilution, Invitrogen) used in this study were as follows: AlexaFluor 488 donkey anti-rabbit antibody (A32790), Alexa-Fluor 594 donkey anti-mouse antibody (A21203), and DyLight 680 goat anti-chicken antibody (SA5-10074). Sections were counterstained with Hoechst (1:10000 dilution, ThermoFisher, H3569) if needed. Slices were stained with H&E following the standard H&E procedure. Images were captured using a confocal microscope (Zeiss 980) and analyzed using ImageJ (NIH).

## TUNEL assay

Terminal deoxynucleotidyl transferase dUTP nick end labeling (TUNEL) staining was performed using One-step TUNEL In Situ Apoptosis Kit (Red, Elab Fluor® 594) (E-CK-A322, Elabscience) following the manual instruction and as we previously described[78]. Images were captured using a confocal microscope (Zeiss 980).

## Food intake measurement

For the rebound feeding experiments, mice were fasted overnight (16 h) with water provided *ad libitum*. The following day, mice were housed individually and provided with a certain weight of food and sufficient water, then the remaining food weight was recorded at designated time points and used to calculate rebound food intake. For the daily food intake experiments, mice were housed individually for 3 consecutive days and provided with a certain weight of food and sufficient water. The remaining food mass was recorded at 10 a.m. each day and used to calculate daily food intake. For exogenous SCT administration experiment, mice were i.p. injected with either SCT (100 μg/kg BW) (067-04, Phoenix Pharmaceuticals) or vehicle (PBS) before refeeding. Mouse lean body mass was analyzed using NMR (LF90 Minispec, Bruker Corp.) before the start of the experiment. Mice were acclimatized for at least 15 min in the cage before the experiments.

## Bone histomorphometry

Bone histomorphometry was performed on both paraffin sections. In brief, the femur specimens, after fixation in 4% PFA for 24 h, were decalcified with 10% ethylenediaminetetraacetic acid (EDTA, Sigma-Aldrich) for 4 weeks. For paraffin sections, the specimens were processed, embedded in paraffin, and cut into 5-μm-thick sections using a rotary microtome (RM215, Leica). H&E staining, TH staining (AB152, Millipore), and TRAP staining (MK30, Takara) were performed on selected sections from each sample following the manufacturer's instructions. Images were captured using the Vectra Polaris Imaging System (Akoya Biosciences) and analyzed using ImageJ (NIH).

## Fluorochrome labeling

Two fluorochrome labels were used sequentially to evaluate bone-formation rate and mineralization. 16-week-old mice were injected subcutaneously with a first dose of calcein (10 mg/kg, C0875, Sigma-Aldrich), followed by a second dose of calcein (10 mg/kg) one week later. The fluorochrome labels were visualized under fluorescence microscopy (Niko ECL IPSE 80i) and analyzed by ImageJ (NIH). The mineral apposition rate (MAR) was calculated by the inter-label width divided by the number of days between label administration.

## In situ hybridization

The mouse was anesthetized with isoflurane and was perfused by PBS solution (RNAase-free). All sample preparation was based on formalin-fixed paraffin-embedded (FFPE) sample preparation and pre-treatment protocols as recommended by RNAscope® Multiplex Fluorescent Assays v2 (323100, Advanced Cell Diagnostics). The *Sct* probe (44999-C1) and the *Nr5a1* probe (445731-C2) were purchased from the Advanced Cell Diagnostics. According to manufacturer's instruction, after removing the parafilm by xylene for 5 min twice, Slides were washed in ethanol for 5 min twice. When slices were dry, hydrogen peroxide was used to remove hydrogen peroxidase of tissue for 15 min. Proteins were digested using protease solution for 15 min in 40 °C. Immediately, slides were washed twice in distilled water for 2 min. In parallel, probes were heated in a 40 °C water bath for 10 min. After warmth, the probe was applied to the slides, which were covered by coverslips and placed in a 40 °C hybridization oven for 3 h. After washing for 3 times in wash buffer, slides underwent the signal amplification process as stipulated in the vendor's protocols (AMP1, AMP2, AMP3). Finally, slices were incubated with Opal-520/Opal-690 staining and were counter-stained by Hoechst. The coverslip was sealed using ProLong™ Diamond Antifade mounting (P36965, ThermoFisher). Images were captured using a confocal microscope (Zeiss 980) and analyzed using ImageJ (NIH).

## μCT analysis

The femurs from each mouse (20-week-old) were dissected and fixed overnight in 4% PFA for 24 h, loaded into 12.3 mm diameter scanning tubes. The femurs were scanned by a high-resolution micro-CT scanner (SkyScan 1276, Bruker) at a resolution of 79.66 μm per pixel. The voltage of the scanning procedure was 70 kV with a 153200-μA current. Two phantom-contained rods with a standard density of 0.25 and 0.75 g/cm³ were used for calibration of bone mineral density (BMD). Data reconstruction was done using the NRecon software (Bruker), the image analysis was done using CTAn software (Bruker), and the 3D model visualization was done using CTvox (Bruker) and CTvol (Bruker). The region of interest (ROI) contained 200 layers of images beginning from the distal metaphyseal growth plate of femurs. Trabecular bone parameters, including bone volume fraction (BV/TV), specific bone surface (BS/BV), bone mineral density (BMD of TV), trabecular thickness (Tb.Th), trabecular number (Tb.N), trabecular pattern factor (Tb.Pf), and trabecular separation (Tb.Sp) were measured from the μCT data.

## Mitochondria contents in the iBAT

For mitochondrial DNA content analysis, total DNA from iBAT was extracted using a FastPure® Blood/Cell/Tissue/Bacteria DNA Isolation Mini Kit (DC112-01, Vazyme biotech) according to the manufacturer's instructions. Mitochondrial DNA was amplified using primers specific for the mitochondrial cytochrome C oxidase subunit 2 (COX2) gene and normalized to genomic DNA by amplification of the 40S ribosomal protein s18 (Rps18) nuclear gene. Primer sequences were listed in Supplementary Table 1.

### RNA isolation and real-time qPCR

Total RNA was extracted from tissue samples (liver, iBAT, and iWAT) using TRIzol™ Reagent (15596026, ThermoFisher) following the manufacturer's instructions; 1 μg of total RNA was used to synthesize cDNA using the HiScript®III All-in-one RT SuperMix (R333-01, Vazyme biotech). An aliquot (1/5 vol) of the cDNA was then subjected to qPCR using the ChamQ SYBR qPCR Master Mix (Q411-02, Vazyme biotech) in a 96-well real-time PCR machine (7300 Real-Time PCR System, Applied Biosystems). Fold changes were calculated and determined using the $2^{-\Delta\Delta Ct}$ method and expression levels normalized to the average of the housekeeping genes 18S. Primer sequences were listed in Supplementary Table 1.

### Western blot

Fresh tissue was rapidly isolated and stored in a −80 °C freezer for subsequent processing. The VMH was separated from the section under a stereo microscope (L-Z2000, Leica). Tissues were homogenized and lysed in RIPA lysis buffer (89900, ThermoFisher) which contains proteinase inhibitors (04693132001, Roche) and phosphatase inhibitors (4906845001, Roche). The lysates were then washed and boiled in SDS loading buffer. Equal amounts of protein lysates were resolved on SDS–polyacrylamide gel electrophoresis gels and transferred to polyvinylidene difluoride membrane. The membranes were blocked in 5% bovine serum albumin and incubated with primary antibodies (1: 5000 dilution): rabbit anti-pCREB (phospho S133) (ab32096, Abcam), mouse anti-CREB (ab178322, Abcam), rabbit anti-TH (AB152, Millipore), rabbit anti-Ucp1 (U6382, Sigma-Aldrich) and rabbit anti-GAPDH (2118, Cell Signalling Technology) overnight at 4 °C. After three washes, the membranes were incubated with peroxidase-conjugated anti-rabbit secondary antibody (7074, Cell Signalling Technology) or peroxidase-conjugated anti-mouse secondary antibody (31430, Invitrogen) for 1 h and visualized with enhanced chemiluminescence substrate (34580, Thermo Scientific). The Bio-Rad ChemiDoc system was used to visualize the blots.

### Quantification of tissue SCT

The whole brain was extracted and sectioned into thick slices (300 μm). The VMH was separated from the section under a stereo microscope (L-Z2000, Leica). Total proteins were extracted using RIPA lysis buffer (89900, ThermoFisher). After protein quantification using the BCA Protein Assay Kit (P0011, Beyotime), the concentration SCT was measured by an ELISA kit (EK-067-04, Phoenix Pharmaceuticals) following the manual instruction.

### Single-cell RNA sequencing (scRNA-seq) data analysis

The single-cell RNAseq data for mouse VMH were obtained from Mendeley Data (https://doi.org/10.17632/ypx3sw2f7c.1). Detailed information for these data can be found in the original paper[49]. The data analysis was performed in R software with Seurat (v4.2.0) package using default parameters unless specified. In brief, scRNA-seq datasets from three different mice were loaded and merged into Seurat. Gene expressions of each cell were log-transformed (NormalizeData function) and highly variable genes were identified (FindVariableGenes function; top 2,000 genes with the highest standardized variance selected by selection.method = 'vst') to be used as input for dimensionality reduction via principal component analysis (PCA). We then used the anchors (FindIntegrationAnchors function) to integrate the datasets together (IntegrateData function), performed a joint clustering on these aligned embeddings (FindClusters function; resolution of 0.8), yielding 33 clusters. To differentiate non-neuronal cells from neuronal cells, marker genes for endothelial cells (*Cldn5*), microglia (*C1qc*), oligodendrocytes (*Opalin*), astrocytes (*Gja1*), oligodendrocyte progenitor cells (*Pdgfra*), mural cells (*Mustn1*) were selected as suggested in the original paper[49]. Neuronal cells were identified by maker gene like *Stmn2* and further categorized into Glutamatergic or GABAergic neurons based on the expression level of *Slc17a6, Fezf1, Adcyap1, Slc32a1*, and *Gad2*.

### Statistics and reproducibility

We performed statistical analyses using Prism software (GraphPad Software, v.7.0). Throughout the paper, values are reported as mean ± SEM (error bars or shaded area). Experiments in this study were repeated independently at least three times. P values for comparisons across two groups were performed using a two-tailed Student's *t*-test. *P* values for comparisons across multiple groups were performed using one- or two-way ANOVA and corrected for multiple comparisons using the Holm–Šídák method. When comparing EE between different groups with BW as a covariate, one-way ANCOVA was performed in R (4.2.2) function aov() based on the formula: Energy expenditure ~ Body weight * Genotypes. Further pairwise comparisons on *adjusted means* were conducted from R package "emmeans", and P values were corrected for multiple comparisons using the Bonferroni method. Significance was defined as *$P < 0.05$, **$P < 0.01$, ***$P < 0.001$. Sample sizes and specific tests are denoted in the figure legends. Details of the statistical analysis are provided in the Source Data files.

### Reporting summary

Further information on research design is available in the Nature Portfolio Reporting Summary linked to this article.

## Data availability

All data generated or analyzed during this study are included in this published article (and its supplementary information files). The single-cell RNAseq data for mouse VMH were obtained from Mendeley Data (https://doi.org/10.17632/ypx3sw2f7c.1), and details of these data can be found in the original article[49]. Source data are provided with this paper.

## Code availability

The customized R code for the ANCOVA analysis is available in GitHub (https://github.com/nedchen2/HKFI-Secretin-Signaling/) and Zenodo (https://doi.org/10.5281/zenodo.10476920).

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

## Acknowledgements

We sincerely thank the staff members at the Faculty Core Facility of Li Ka Shing Faculty of Medicine and the Centre for Comparative Medicine Research at the University of Hong Kong, for their facilitation of the study. The work is supported by funding from the Research Grants Council, the Government of the Hong Kong SAR (General Research Fund Nos. 17126222, 17113120, and 17115923 to B.K.C.C.; Nos. 17111322, 17207719 and 17214516 to K.W.K.Y.; Collaborative Research Fund No. C7003-22Y to W.Q.; No. C5044-21GF to K.W.K.Y.); Seed Funding for Strategic Interdisciplinary Research Scheme to B.K.C.C.; the Food and Health Bureau, the Government of the Hong Kong SAR (No. 09201466 to W.Q.; No. 19180712, No. 20190422, and No. 21200592 to K.W.K.Y.); National Natural Science Foundation of China (Young Scientist Fund No.82201124 to W.Q.); National Natural Science Foundation of China / Research Grants Council Joint Research Scheme (N_HKU721/23 to W.Q.); Hong Kong Innovation Technology Fund (ITS/405/18 to K.W.K.Y., ITS/256/22 to W.Q.); National Key R&D Program of China (2018YFA0703100 to K.W.K.Y.); Shenzhen Science and Technology Funding (No.JCYJ20210324120012034 and No.JCYJ20210324120009026 to K.W.K.Y.), Shenzhen Science and Technology Innovation Committee Projects (Nos. SGDX20220530111405038, W.Q.), Guangdong Basic and Applied Basic Research Foundation (2023A1515011963, W.Q.) and HKU-SZH Fund for Shenzhen Key Medical Discipline (SZXK2020084 to K.W.K.Y.).

## Author contributions

F.Z. and W.Q. designed the study. F.Z. prepared animal models. F.Z. and W.Q. performed experiments. J.W. and L.Z. performed the in situ hybridization experiment. F.Z., Z.T., Y.W., and M.L. performed the western blot experiment. F.Z., W.Q., and K.M.C.N. performed the histological experiment. F.Z. and W.Q. prepared figures. F.Z., W.Q., and C.C. analyzed and interpreted the data. F.Z., and W.Q. wrote the manuscript, with input from all authors. W.Q., K.W.K.Y., and B.K.C.C. provided fundings and supervised the study.

## Competing interests

The authors declare no competing interests.
