## [Peer Review File · Nature Communications]

REVIEWER COMMENTS

Reviewer #1 (Remarks to the Author):

In the article by Zhang et al, the authors investigated the function of the secretin signalling system in terms of whole-body bone and energy homeostasis. They use a comprehensive approach utilising a battery of genetic mouse models and find that specifically secretin derived from the VMH in the hypothalamus is critical for the regulation of bone as well as energy metabolism. Specifically, they identify that lack of secretin signalling in the VMH leads to an increase in food intake which also develops into an obese phenotype as well as exhibits reduced bone mass. They link these effects to altered POMC neuron activity as well as changes in SNS outflow.

The experiments appear to have been performed with care and the data analysis being conducted with the appropriate statistics. The paper in general is also well written and provides an objective discussion of the results.

However, I do have a few questions and comments regarding some of the experiments and data.

With regards to the observed down-regulation of the POMC mRNA are there actually secretin receptors being located on POMC neurons? How do secretin receptors signal, inhibitory or stimulatory and is secretin signalling the main driver of the downregulation of POMC mRNA.

Is the effect on these neurons direct or indirect or mainly due to glutamate signalling rather than secretin?

The data on thermogenesis are interesting. Have the data presented for UCP-1 and other markers been normalised to tissue size/weight? While down-regulated in the KO models due to the increase in size/mass of BAT in these models this could just be a dilution effect using an aliquot, but the absolute total amount of UCP-1 produced in the depot might not actually be different between the genotypes.

With the bone analysis I notice a large difference in basal BV/TV levels in WT/controls between the different experiments eg 20 in Fig 1I and only 10 in Fig 7F. How do the authors explain such a 'drifting' baseline and how does this effect the result/interpretation when for example the over expression of secretin in Fig 7F produces an increase in BV/TV that in absolute value is not much different from the KO model (Fig 1 I).

Have female mice been tested as well? Is there any evidence that both genders behave the same?

Reviewer #2 (Remarks to the Author):

The authors reproduced previous findings with respect to the association of systemic SCT or SCTR deficiency with increased food intake without body weight gain. They found an upregulation of sympathetic nerve activity. They also reproduced previous findings suggesting that systemic SCT or SCTR deficiency led to a significant decrease in bone mineral density.

In a series of experiments, they provide new evidence for an important role for specific VMH-SCT in the regulation of sympathetic nerve activity and bone mineral density. Interestingly, they also provide new evidence for VMH-SCT in the regulation of appetite and subsequent obesity.

These findings are of interest. Ultimately, VMH-SCT may serve as a new therapeutic target for the clinical management of obesity and osteoporosis. However, the animal findings are very basic and far from clinical practice.

Are the animal models relevant for humans? How do the authors think their findings will have consequences for clinical practice? Are there examples of interventions that can be used to influence the hypothalamus in humans? What do they think should be the next steps?

Reviewer #3 (Remarks to the Author):

This manuscript reports on the role of SCT and SCTR in hypothalamic VMH of mice. The experiments reveal a role of central SCT signalling in the control of bone formation, namely the balance of osteoblast and osteoclast activity and the formation of bone marrow adipocytes. Ablation of SCT signaling in the VMH led to increased sympathetic output which is known to inhibit bone formation. In addition, SCT signaling in VMH affected food intake, energy expenditure, activity behaviour and energy metabolism. The results obtained were reproducible in different mouse models with germline or conditional SCT or SCTR ablation, either using shRNA or Cre-mediated knockdown, but also SCT overexpression with AAV vectors.

This study appears to be of high quality and provides new insights. In particular, the role of SCT signaling in VMH for bone health is novel and will attract considerable interest.

Several points need to be addressed:

Align the outline of the abstract with the other sections of the manuscript, or vice versa.

A limitation of the study is that the known negative effects of adipokines on bone health cannot be distinguished from the effect triggered by SCT ablation in VMH.

Could it be that SCT acts as a neurotrophic factor in the VMH? With SCTR expressed in the VMH, SCT ablation could affect viability of VMH neurons. Perhaps SCT ablation induces cell death. Can you exclude that the phenotype observed is induced by a loss of VMH neuron function, and thereby phenocopies the classical lesion experiments.

Are the central and peripheral effects of SCT on energy metabolism separated, or do they depend on each other? For example, does peripheral SCT injection inhibit feeding in mice with VMH specific SCT ablation?

All food intake and energy expenditure data are normalized by body mass. With considerable differences in body mass and body composition, ANCOVA based analysis or lean mass-specific ratios should be calculated and subjected to statistical testing. Data must be reanalyzed accordingly. For example, in Figure 4, EE is lower in shSCT, but is this simply because you divide by a large body mass, or is body mass / composition adjusted EE also lower? There are several technical reviews on this topic that should be consulted to apply the proper method for the analysis.

Increased levels of NE in blood or tissues only are a surrogate measure of increased sympathetic tone.

Line 348: what is the evidence for bulimia? Hyperphagia would be the more appropriate term.

Line 351 ff: Please clarify this discussion section. Why should it be worthwhile looking into SCTR signaling in POMC neurons? Do strong excitatory neuronal projections from VMH regulate POMC neuron activity in the ARC using SCT as the neuropeptidergic neurotransmitter? Excitatory neurons should be glutamatergic. Or are you suggesting that SCT acts in a paracrine mode diffusing from VMH to ARC to activate SCTR in POMC neurons?

The new data on food intake and body mass regulation in global SCT and SCTR knockout mice fed regular chow diet show genotype effects that were not seen in the previous work. This should be mentioned / discussed.

Sctr and Sct must be added to extended Figure 4C.

Line 238: TH is an enzyme, not a NE precursor.

Line 240: improve the wording: The impaired activity or function or tone (instead of innervation) of TH+ sympathetic nerves

The Discussion should end with a paragraph summarizing the main findings and presenting a perspective.

Reviewer #1:

In the article by Zhang et al, the authors investigated the function of the secretin signalling system in terms of whole-body bone and energy homeostasis. The use a comprehensive approach utilizing a battery of genetic mouse models and find that specifically secretin derived from the VHM in the hypothalamus is critical for the regulation of bone as well as energy metabolism. Specifically, the identify that lack of secretin signalling in the VMH leads to an increase in food intake which also develops into an obese phenotype as well as exhibits reduced bone mass. They link these effects to altered POMC neuron activity as well as changes in SNS outflow.

The experiments appear to have been performed with care and the data analysis being conducted with the appropriate statistics. The paper in general is also well written and provides an objective discussion of the results.

We thank the reviewers for the encouraging comments.

However, I do have a few questions and comments regarding some of the experiments and data.

1. With regards to the observed down-regulation of the POMC mRNA are there actually secretin receptor being located on POMC neurons? How do secretin receptor signal, inhibitory or stimulatory and is secretin signalling the main driver of the downregulation of POMC mRNA.

Eg is the effect on these neurons direct or indirect or mainly due to glutamate signalling rather than secretin?

We thank the reviewers for raising this issue. To address the question concerning whether SCT directly works on POMC neurons through SCTR, we generated SCTR-Cre::ROSA-tdTomato mice to visualize the expression of SCTR in the central nervous system¹. We showed tdTomato signal was barely detected in the ARC cell body (see below Reply1), indicating the absence of SCTR in ARC^{POMC} neurons. We hypothesize that SCT signaling in the VMH affects POMC expression in the downstream ARC by regulating VMH glutamatergic output. Previous studies have demonstrated that the intense glutamatergic projections of VMH neurons to ARC^{POMC} neurons². The downregulation of glutamatergic output from VMH led to decreased POMC mRNA expression in ARC^{POMC} neurons^{3, 4}. In our study, specific knockdown of SCT in VMH reduces the phosphorylation level of CREB in VMH neurons, which indicates the increased neuronal activity⁵ and the expression of glutamate transporter⁶.

Overall, current evidence supports that changes in ARC^{POMC} neurons are due to glutamate signaling from VMH^{SCT} neurons rather than the direct ligand-receptor effect of SCT. We are currently conducting a new study aim at investigating the

detailed mechanisms of SCT signaling in the VMH^{SCT}-ARC^{POMC} microcircuit, including electrophysiological studies investigating glutamatergic input received by ARC^{POMC} neurons.

Reply 1. Representative images of the ARC of SCTR-Cre::ROSA-tdTomato double transgenic mice. Scale bars = 200 μ m.

References

- 1 Liu Y, et al. A gut-brain axis mediates sodium appetite via gastrointestinal peptide regulation on a medulla-hypothalamic circuit. *Sci Adv.* 2023 Feb 15;9(7).
- 2 Sternson SM, et al. Topographic mapping of VMH  arcuate nucleus microcircuits and their reorganization by fasting. *Nat Neurosci.* 2005 Oct;8(10):1356-63. doi: 10.1038/nn1550.
- 3 Rau AR, et al. Energy state alters regulation of proopiomelanocortin neurons by glutamatergic ventromedial hypothalamus neurons: pre- and postsynaptic mechanisms. *J Neurophysiol.* 2021 Mar 1;125(3):720-730. doi: 10.1152/jn.00359.2020.
- 4 Sun JS, et al. Ventromedial hypothalamic primary cilia control energy and skeletal homeostasis. *J Clin Invest.* 2021 Jan 4;131(1):e138107. doi: 10.1172/JCI138107.
- 5 Moore AN, et al. Neuronal activity increases the phosphorylation of the transcription factor cAMP response element-binding protein (CREB) in rat hippocampus and cortex. *J Biol Chem.* 1996 Jun 14;271(24):14214-20. doi: 10.1074/jbc.271.24.14214.
- 6 Karki P, et al. cAMP response element-binding protein (CREB) and nuclear factor κ B mediate the tamoxifen-induced up-regulation of glutamate transporter 1 (GLT-1) in rat astrocytes. *J Biol Chem.* 2013 Oct 4;288(40):28975-86. doi: 10.1074/jbc.M113.483826.

2. The data on thermogenesis are interesting. Have the data presented for UCP-1 and other markers been normalised to tissue size/weight? While down-regulated in the KO models due to the increase size/mass of BAT in these models this could just be a dilution effect using an aliquot, but the absolute

total amount of UCP-1 produced in the depot might not actually be different between the genotypes.

We thank the reviewer for raising these points. In this study, we utilized the $2^{-\Delta\Delta CT}$ method to quantitatively analyze the gene expression of *Ucp1* and other markers in BAT using *18S* as the housekeeping gene, therefore, it is not likely the data will be biased by the tissue weight. However, we agree with the reviewer that it is necessary to know the absolute content of Ucp1 in BAT. Therefore, in our revised manuscript, we supplemented the Ucp1 levels in the overall iBAT depot determined by ELISA (see below Reply 2 or Fig. 4g and Supplementary Fig. 6k, l). We showed that the knockdown of SCT in VMH led to increased tissue weight of BAT (Reply 2a) and reduced protein density (Reply 2b) in the iBAT. This phenotype is similar to the diet-fed monogenic obesity models, including *ob/ob* mice¹ and Zücker diabetic fatty rats². We also showed that the concentration (normalized to the total protein) and the total amount of Ucp1 in ShSCT iBAT were reduced compared with the control (Reply 2c, d), which is consistent with the qPCR result in this study.

For the method used, please see below or the Methods Line 663-668.

“The Ucp1 concentration was measured using an ELISA kit (SEF557Ra, Cloud-Clone Corp.) following the manual instruction. The total Ucp1 content of iBAT depot was calculated as iBAT depot mass * iBAT protein density * Ucp1 concentration. Protein density was measured using the BCA Protein Assay Kit (P0011, Beyotime).”

Reply 2. **a**, Tissue weight of iBAT. **b**, Protein density of iBAT. **c**, The concentration of Ucp1 protein in iBAT. **d**, Total amount of Ucp1 protein in iBAT depot. Two-tailed Student's *t*-test. ***P* < 0.01; ****P* < 0.001. Error bars represent SEM.

References

1 Fischer AW, et al. *Leptin Raises Defended Body Temperature without Activating Thermogenesis. Cell Rep.* 2016 Feb 23;14(7):1621-1631. doi: 10.1016/j.celrep.2016.01.041.

2 Fernández Vázquez G, et al. *Melatonin increases brown adipose tissue mass and function in Zucker diabetic fatty rats: implications for obesity control. J Pineal Res.* 2018 May;64(4):e12472. doi: 10.1111/jpi.12472.

3. With the bone analysis I notice a large difference in basal BV/TV levels in WT/controls between the different experiments eg 20 in Fig 1I and only 10 in Fig 7F. How do the authors explain such a ‘drifting’ baseline and how does this effect the result/interpretation when for example the over expression of secretin in Fig 7F produces an increase in BV/TV that in absolute value is not much different from the KO model (Fig 1 I).

We thank the reviewers for raising these concerns. We believe this may be due to background differences in the mice used in these two experiments. The mice used in Fig. 7f were animals purchased from the animal breeding center of the University at 4 weeks old. While the systemic KO and WT mice in Fig. 1i were bred in our laboratory. The discrepancies in genetic backgrounds and rearing conditions could contribute to the observed drift in the baseline bone phenotype. Additionally, it is important to consider the long-term adaptive responses due to genetic ablation and the potential negative effects associated with viral injection surgery.

To parallelly compare the influence of SCT knockdown and upregulation on bone homeostasis, we suggest looking at bone phenotypes of VMH-specific SCT knockdown in Fig. 2f and the VMH-SCT overexpression in Fig.7f, in which the mice involved are of the same background. Specifically, with the trabecular bone fraction located at a range of 10-14 in both control groups, the downregulation of SCT in VMH decreased the BV/TV to 5-7, while the upregulation of SCT in VMH increased the BV/TV to 15-20. To further confirm the effects of SCT upregulation on bone homeostasis, we repeated the experiments in male mice and achieved consistent data, which has been included in revised Fig. 7f. Additionally, we also reproduced the bone-promoting effect of SCT overexpression in female mice to validate our findings (Supplementary Fig. 14).

4. Have female mice being tested as well? Is there any evidence that both genders behave the same?

We thank the reviewer for raising this point. We first confirmed the expression of SCT in the VMH of female mice by RNAscope (see below Reply 3 or Supplementary Fig.4a). Next, we generated female mice with systemic SCT or SCTR KO and demonstrated consistent findings on male mice regarding the effects of SCT on body weight, food intake, and bone homeostasis (Supplementary Fig.1). Similarly, VMH-

specific SCT knockdown and VMH-SCT overexpression induced the same effects in female mice as that observed in male (Supplementary Fig.4, 9, 14). Therefore, our data convincingly showed that the function of SCT is not gender dependent.

Reply 3. RNAscope *in situ* hybridization of *Sct* (green) and *Nr5a1* (magenta) in the VMH sections of female mice. Scale bars = 50 μ m.

Reviewer #2 (Remarks to the Author):

The authors reproduced previous findings with respect to the association of systemic SCT or SCTR deficiency with increased food intake without body weight gain. They found an upregulation of sympathetic nerve activity. They also reproduced previous finding suggesting that systemic SCT or SCTR deficiency led to a significant decrease in bone mineral density.

In a series of experiments, they provide new evidence for an important role for specific VMH-SCT in the regulation of sympathetic nerve activity and bone mineral density. Interestingly, they also provide new evidence for VMH-SCT in the regulation of appetite and subsequent obesity.

These findings are of interest. Ultimately, VMH-SCT may serve as a new therapeutic target for the clinical management of obesity and osteoporosis. However, the animal findings are very basic and far from clinical practice.

We thank the reviewers for the encouraging comments and the valuable suggestions. We apologize for any confusion and would like to clarify that our study is indeed the first to report the effect of SCT/SCTR on bone homeostasis. In particular, we uncovered the previously unknown role of secretin in VMH for the control of energy metabolism. Therefore, our finding is meaningful not only in terms of basic science but also hold significant translational potential.

Are the animal models relevant for humans?

We thank the reviewer for this insightful comment. We acknowledge that our experiments are conducted using a mouse animal model due to the inherent difficulties associated with conducting similar investigations in human subjects. Nevertheless, it's important to note that sequence comparison has revealed secretin family to be highly conserved in vertebrate that the mouse secretin is more than 85% identical in amino acid sequence with the human homologues¹. Moreover, human secretin analog has been shown to function effectively in mice², indicating the conserved nature of secretin receptor.

To further verify our finding regarding the effect of secretin signaling in VMH, we reanalyzed the single-cell RNA sequencing data of VMH specimen from rhesus monkeys³ (see below Reply 4). Our analysis revealed a consistent expression pattern of SCT and SCTR in primates, similar to that observed in rodents. This indicates that the SCT-SCTR axis in VMH should be evolutionarily conserved and likely functions similarly among different species, including humans.

Reply 4. The expressions of *Sctr*, *Sct*, and marker genes for astrocytes (*Gja1*) and glutamatergic neurons (*Slc17a6*) of VMH from rhesus monkeys are color coded (Red) on UMAP plots. Raw data were obtained from NCBI Gene Expression Omnibus ID GSE172207.

How do the authors think their findings will have consequences for clinical practice?

Due to the similarity of SCT signaling in rodents and primates, we believe our exciting findings on the therapeutic effect of VMH SCT signaling would hold significant clinical translational potential. In fact, the relevance between secretin and energy/ bone homeostasis has been reported in several clinical studies involving human subjects. For example, a clinical trial found that women with normal bone mass had significantly higher secretin level than with osteopenia. Thus, decrease in secretin may serve as an early marker for the diagnosis of type I osteoporosis in postmenopausal women⁴.

Another clinical study conducted on healthy normal weight men showed that the use of secretin can activate the BAT-brain axis, leading to reduced central responses to appetizing food and delays the motivation to refeed after a meal⁵. Therefore, it was proposed that secretin can be used to treat obesity in clinical practice due to its effects on the regulation of energy metabolism.

Here in this study, our findings deepen our understanding on the underlying mechanism contributing to the impact of secretin on bone homeostasis and energy metabolism. Specifically, VMH^{SCT}-ARC^{POMC} circuit could be used as the target for the development of drugs to treat appetite disorders or obesity. Meanwhile, VMH^{SCT}-SNS pathway could be used as a novel approach for the management of osteoporosis.

Are there examples of interventions that can be used to influence the hypothalamus in humans?

We sincerely thank the reviewer for raising this important point. In this study, we used stereotaxic injection of AAV to specifically modulate VMH SCT signaling, which is not very applicable to patients in clinical practice. However, there have been many non-invasive ways to delivery therapeutic agents to the central nervous system. For example, Intranasal administration of nanoparticles, liposomes and polymeric micelles have been shown able to effectively deliver brain targeting drugs in clinical practice⁶. Moreover, extracellular exosomes capable of crossing BBB can be engineered for targeted delivering of small-molecule drugs, nucleic acids and proteins to the central nervous system⁷.

A clinical trial in human subjects using functional magnetic resonance showed that peripheral infusion of secretin resulted in improved inhibitory control in several brain regions and thereby, downregulated the brain response to appetizing food images⁸. Therefore, this study reveals the potential of peripherally administrated secretin to function in CNS of human.

What do they think should be the next steps?

We thank the reviewer for the valuable comment. With the discovery of the central role of SCT signaling in the control of energy metabolism and bone homeostasis, we aim to explore the best approach to activate SCTR in VMH for the treatment of obesity or osteoporosis in both animal study and clinical trial.

For instance, we are currently testing the efficacy of a small molecule, KSD179019, which is a SCTR positive allosteric modulator that has passed acute, sub-chronic, and chronic toxicology evaluations⁹. So far, our preliminary data is promising, as acute administration of KSD179019 has been found to enhance thermogenesis and reduce appetite after fasting. Moreover, long-term oral administration of KSD179019

can effectively reduce body weight in mice with diet-induced obesity. Therefore, in the next step, we will further test it in clinical trials and commercialize the product to be used in clinical practice. In addition, we are also working with our collaborators to develop nano-sized biomaterials specifically designed for targeted delivery of KSD179019 to the VMH. This approach aims to enhance the specificity of the treatment while minimizing potential side effects associated with systemic administration.

References

1. Cardoso, João CR, et al. "The serendipitous origin of chordate secretin peptide family members." *BMC evolutionary biology* 10.1 (2010): 1-19.
- 2 Laurila S, et al. Secretin activates brown fat and induces satiation. *Nat Metab.* 2021 Jun;3(6):798-809. doi: 10.1038/s42255-021-00409-4. Epub 2021 Jun 21.
- 3 Affinati AH, et al. Cross-species analysis defines the conservation of anatomically segregated VMH neuron populations. *Elife.* 2021 May 21;10:e69065.
- 4 He WT, et al. Weak cation exchange magnetic beads coupled with matrix-assisted laser desorption ionization-time of flight-mass spectrometry in screening serum protein markers in osteopenia. *Springerplus.* 2016 May 21;5(1):679. doi: 10.1186/s40064-016-2276-4.
- 5 Laurila S, et al. Secretin activates brown fat and induces satiation. *Nat Metab.* 2021 Jun;3(6):798-809. doi: 10.1038/s42255-021-00409-4. Epub 2021 Jun 21.
- 6 Khan, Abdur Rauf, et al. "Progress in brain targeting drug delivery system by nasal route." *Journal of Controlled Release* 268 (2017): 364-389.
- 7 Gratpain, Viridiane, et al. "Extracellular vesicles for the treatment of central nervous system diseases." *Advanced Drug Delivery Reviews* 174 (2021): 535-552.
- 8 Sun L, et al. Secretin modulates appetite via brown adipose tissue-brain axis. *Eur J Nucl Med Mol Imaging.* 2023 May;50(6):1597-1606. doi: 10.1007/s00259-023-06124-4. Epub 2023 Feb 11.
- 9 S.A. Nawabjan, et al. LP-35 Sub-chronic and Chronic toxicity study of Investigational Anti-Hypertensive Small Molecule KSD179019. *Toxicology Letters.* 2022, Page S296. doi.org/10.1016/j.toxlet.2022.07.775.

Reviewer #3 (Remarks to the Author):

This manuscript reports on the role of SCT and SCTR in hypothalamic VMH of mice. The experiments reveal a role of central SCT signalling in the control of bone formation, namely the balance of osteoblast and osteoclast activity and the formation of bone marrow adipocytes. Ablation of SCT signaling in the VMH led to increased sympathetic output which is known to inhibit bone formation. In addition, SCT signaling in VMH affected food intake, energy expenditure, activity behaviour and energy metabolism. The results obtained were reproducible in different mouse models with germline or conditional SCT or SCTR ablation, either using shRNA or Cre-mediated knockdown, but also SCT overexpression with AAV vectors.

This study appears to be of high quality and provides new insights. In particular, the role of SCT signaling in VMH for bone health is novel and will attract considerable interest.

We appreciate the reviewer's encouraging comments.

Several points need to be addressed:

Align the outline of the abstract with the other sections of the manuscript, or vice versa.

We thank the reviewer for the suggestion and have carefully formatted the manuscript accordingly.

A limitation of the study is that the known negative effects of adipokines on bone health cannot be distinguished from the effect triggered by SCT ablation in VMH.

We agree with the reviewer that the obese phenotype induced by VMH-SCT ablation may upregulate adipokine, leading to exacerbated negative effects on bone health. This is not surprising as, leptin, one of the most extensively studied adipokines that is involved in the crosstalk with bone, has been known to suppress bone formation through its regulatory effect in hypothalamus¹ and direct effect on bone tissue².

However, it is extremely challenging to exclude the involvement of adipokine *in vivo*. Therefore, we sought to determine the plasma leptin levels after the conditional knockdown of SCT or SCTR in VMH and relate it to the bone phenotype. We showed that although the ablation of SCT or SCTR led to increased leptin in chow-fed mice following the induction of obesity, there was no significant change in leptin levels in VMH SCT knockdown mice compared to control mice when they were fed with HFD (see below Reply 5a or Fig. 6c). Nevertheless, a significant decrease in bone density was still evident in the knockdown group relative to the control group (Reply 5b or Fig. 6j). Therefore, our data suggests that the alteration in bone

phenotype induced by the loss of VMH SCT should not be entirely adipokine-dependent.

Reply 5. **a**, Serum leptin levels in HFD-fed SCT^{VMH}^{-/-} and eGFP littermates. **b**, BV/TV, BMD of TV, and Tb.N of femurs from 20-week-old HFD-fed SCT^{VMH}^{-/-} and eGFP littermates. Two-tailed Student's *t*-test. Error bars represent SEM.

References

- 1 Takeda S, et al. Leptin regulates bone formation via the sympathetic nervous system. *Cell*. 2002 Nov 1;111(3):305-17. doi: 10.1016/s0092-8674(02)01049-8.
- 2 Turner RT, et al. Peripheral leptin regulates bone formation. *J Bone Miner Res*. 2013 Jan;28(1):22-34. doi: 10.1002/jbmr.1734.

Could it be that SCT acts as a neurotrophic factor in the VMH? With SCTR expressed in the VMH, SCT ablation could affect viability of VMH neurons. Perhaps SCT ablation induces cell death. Can you exclude that the phenotype observed is induced by a loss of VMH neuron function, and thereby phenocopies the classical lesion experiments.

We appreciate this important issue raised by the reviewer. Indeed, some studies reported the function of SCT in maintaining cell survival and synaptic plasticity in the cerebellum¹. However, it remains unknown whether SCT functions similarly in the hypothalamus.

To address this question, we compared the expression of apoptosis marker Cleaved Caspase-3 in the VMH of ShSCT and ShCon mice and showed there is no difference between these two groups (please see below **Reply 6a** or **Supplementary Fig. 5a**). To verify our finding, we also used TUNEL staining to detect VMH cell death. There was no difference in cell viability between the ShSCT and ShCon

groups. (Reply 6a or Supplementary Fig. 5a). At last, we further studied the number of neurons in VMH through the immunofluorescent staining of NeuN, which serves as a neuronal marker. We showed that there was no difference in neuronal density in VMH between ShSCT and ShCon mice (Reply 6b, c or Supplementary Fig. 5b, c). These results consistently confirmed that SCT knockdown in VMH did not induce apoptosis of VMH cells. Therefore, we propose that the phenotypes induced by SCT ablation in VMH were not caused by the changes in the viability of VMH cells. Consequently, we can conclude that the SCT ablation-induced modulation in VMH is not a simple phenocopy of VMH lesion experiments.

Reply 6. VMH-specific SCT KD did not affect cell survival and neuronal density in the VMH. **a**, Immunofluorescence staining of apoptosis markers (TUNEL and Cleaved Caspase-3) in VMH of ShSCT and ShCon littermates. **b**, Immunofluorescence staining of NeuN in VMH of ShSCT and ShCon mice. **c**, Neuronal density calculated based on NeuN staining (**b**). eGFP indicates the area of virus injection. Numbers in parentheses in each graph indicate sample size. Two-tailed Student's *t*-test. *Ns*, no significant. Error bars represent SEM. Scale bar = 100 μm .

References

1 Wang L, Zhang L. Involvement of Secretin in the Control of Cell Survival and Synaptic Plasticity in the Central Nervous System. *Front Neurosci.* 2020 May 6;14:387. doi: 10.3389/fnins.2020.00387.

Are the central and peripheral effects of SCT on energy metabolism separated, or do they depend on each other? For example, does peripheral SCT injection inhibit feeding in mice with VMH specific SCT ablation?

We thank the reviewer for these important questions. Based on existing evidence, it is likely that central and peripheral SCT regulate energy metabolism through different mechanisms. Our study shows that central SCT affects energy metabolism through the VMH-SNS pathway and regulates appetite through the VMH^{SCT}-ARC^{POMC} pathway. On the other way, peripheral SCT mediates postprandial satiation and thermogenesis through the gut-BAT-brain axis¹. In addition, peripheral SCT is also involved in intestinal lipid absorption through its effects on jejunal cells².

While SCT has previously been suggested as capable of crossing the blood–brain barrier³, in our study, the phenotypes induced by VMH-specific SCT ablation indicate the lack of SCT in CNS can not be rescued by peripheral SCT. To further test our hypothesis, we performed additional experiments to study the effects of peripheral administration of SCT on satiation. In both ShCon and ShRNA mice, additional peripheral SCT contributed to significantly reduced food intake. However, in ShRNA mice, peripheral supplementation of SCT failed to restore the food intake back to the normal level observed in ShCon mice (please see below Reply 7 or Fig. 3c).

These new data suggest that appetite is regulated by SCT through multiple pathways in parallel. It will then be interesting to further study how central and peripheral SCT are coordinated to simultaneously contribute to the control of satiation.

Reply 7. Rebound cumulative (left) and total (right) food intake after SCT administration in overnight fasted ShSCT and ShCon littermates. Mice were injected (i.p.) with either SCT (100 μ g/kg BW) or vehicle (PBS) before refeeding. Two-way ANOVA with Holm–Šídák multiple comparisons test. *P < 0.05; ***P < 0.001. Error bars represent SEM.

References

- 1 Li Y, et al. Secretin-Activated Brown Fat Mediates Prandial Thermogenesis to Induce Satiation. *Cell*. 2018 Nov 29;175(6):1561-1574.e12. doi: 10.1016/j.cell.2018.10.016.
- 2 Sekar R et al. Secretin receptor-knockout mice are resistant to high-fat diet-induced obesity and exhibit impaired intestinal lipid absorption. *FASEB J*. 2014 Aug;28(8):3494-505. doi: 10.1096/fj.13-247536.
- 3 Banks WA, et al. Differential transport of a secretin analog across the blood-brain and blood-cerebrospinal fluid barriers of the mouse. *J Pharmacol Exp Ther*. 2002 Sep;302(3):1062-9. doi: 10.1124/jpet.102.036129.

All food intake and energy expenditure data are normalized by body mass. With considerable differences in body mass and body composition, ANCOVA based analysis or lean mass-specific ratios should be calculated and subjected to statistical testing. Data must be reanalyzed accordingly. For example, in Figure 4, EE is lower in shSCT, but is this simply because you divide by a large body mass, or is body mass / composition adjusted EE also lower? There are several technical reviews on this topic that should be consulted to apply the proper method for the analysis.

We sincerely appreciate the valuable suggestion from the reviewer and have reanalyzed the food intake and energy expenditure data.

Regarding the energy expenditure results, we have adopted the ANCOVA analysis method based on recommendations from a technical review titled "A guide to analysis of mouse energy metabolism"¹ and several surrounding papers²⁻⁵. According to these studies, ANCOVA or generalized linear modeling is the most appropriate statistical approach to accommodate discrete (genotype) and continuous (body mass) traits, rather than using a simple division by BW or lean BW. Therefore, reanalysis will prevent overcompensation of mass effects by BW or lean BW. Reanalyzed data showed that VMH-specific SCT knockdown resulted in decreased energy expenditure in mice fed on chow diet and HFD, please see Fig. 4b, 6g and Supplementary Fig. 6i, 12g, 13g. For the details of ANCOVA analysis method, please see the Methods Line 833-837.

For the food intake results, because our and previous studies⁶ show that mouse food intake is non-linearly related to body weight, the conditions for applying the ANCOVA were not met. We therefore normalized food intake results based on lean BW. Reanalysis showed that systemic knockout or VMH-specific SCT knockdown resulted in increased food intake in mice fed on conventional diet and HFD, please see Fig. 1a-c, 2a-c, 2o, 2p, 5i, 5j, 6d and Supplementary Fig. 1a-c, 9a, 9b, 13e.

References

- 1, Tschöp, M., Speakman, J., Arch, J. et al. A guide to analysis of mouse energy metabolism. *Nat Methods* 9, 57–63 (2012).
2. Allison DB, Paultre F, Goran MI, Poehlman ET, Heymsfield SB. Statistical considerations regarding the use of ratios to adjust data. *Int. J. Obes.* 1995;19:644–652.
3. Poehlman ET, Toth MJ. Mathematical ratios lead to spurious conclusions regarding age-related and sex-related differences in resting metabolic-rate. *Am. J. Clin. Nutr.* 1995;61:482–485.
4. Ravussin E, Bogardus C. Relationship of genetics, age, and physical fitness to daily energy-expenditure and fuel utilization. *Am. J. Clin. Nutr.* 1989;49:968–975.
5. Schmidt MV, et al. A novel chronic social stress paradigm in female mice. *Horm. Behav.* 2010;57:415–420.
6. Yang Y, et al. Variations in body weight, food intake and body composition after long-term high-fat diet feeding in C57BL/6J mice. *Obesity (Silver Spring)*. 2014 Oct;22(10):2147-55.

Increased levels of NE in blood or tissues only are a surrogate measure of increased sympathetic tone.

We agree with this point raised by the reviewer. To further verify our argument regarding the increased sympathetic tone, we further measured TH levels in bones. TH is the rate-limiting enzyme in NE biosynthesis¹, and its activity is a key index of sympathetic tone, reflecting preganglionic neuron activity and the functional integrity of synapses on sympathetic neurons². Thus, the level of TH can reflect the sympathetic innervation of the specific tissue. Our immunofluorescence results show that the TH signal intensity of sympathetic nerves is upregulated in the bone marrow of shSCT mice, which is consistent with the findings based on blood NE levels, indicating an enhanced sympathetic output resulting from VMH-SCT knockdown, please see below Reply 8 or Supplementary Fig. 8.

Reply 8. VMH-specific SCT KD enhances sympathetic output in bone tissue. a, Representative immunofluorescent images showing the presence of TH positive sympathetic nerves in bone marrow of ShSCT and ShCon littermates. Arrows indicate TH-positive signals. Scale bar = 500 μ m. **b,** Quantitative analysis of (a). Numbers in parentheses in each graph indicate sample size. Two-tailed Student's *t*-test. ** $P < 0.01$. Error bars represent SEM.

References

1. Nagatsu T, Levitt M, Udenfriend S: Tyrosine hydroxylase: the initial step in norepinephrine biosynthesis. *J Biol Chem* 239:2910-17, 1964
2. Chalazonitis A, Zigmond RE: Effects of synaptic and antidromic stimulation on tyrosine hydroxylase activity in the rat superior cervical ganglion. *J Physiol* 300:525-36, 1980

Line 348: what is the evidence for bulimia? Hyperphagia would be the more appropriate term.

We agree with this and have replaced “bulimia” with “hyperphagia”, please also see Line 366.

Line 351 ff: Please clarify this discussion section. Why should it be worthwhile looking into SCTR signaling in POMC neurons? Do strong excitatory neuronal projections from VMH regulate POMC neuron activity in the ARC using SCT as the neuropeptidergic neurotransmitter? Excitatory neurons should be glutamatergic. Or are you suggesting that SCT acts in a paracrine mode diffusing from VMH to ARC to activate SCTR in POMC neurons?

We apologize for the confusion and thank the reviewers for raising this issue. We have revised the discussion related to this, please see the Discussion, Line 374-379.

To address the question concerning whether SCT directly works on POMC neurons through SCTR, we recently generated SCTR-Cre::ROSA-tdTomato mice to visualize the expression of SCTR in the central nervous system¹. We showed tdTomato signal was barely detected in the ARC cell body (see below Reply 9), indicating the absence of SCTR in ARC^{POMC} neurons. We hypothesize that SCT signaling in the VMH affects POMC expression in the downstream ARC by regulating VMH glutamatergic output. Previous studies have demonstrated that the intense glutamatergic projections of VMH neurons to ARC^{POMC} neurons². The downregulation of glutamatergic output from VMH led to decreased POMC mRNA expression in ARC^{POMC} neurons^{3, 4}. In our study, specific knockdown of SCT in VMH reduces the phosphorylation level of CREB in VMH neurons, which indicates the increased neuronal activity⁵ and the expression of glutamate transporter⁶.

Overall, current evidence supports that changes in ARC^{POMC} neurons are due to glutamate signaling from VMH^{SCT} neurons rather than the direct paracrine effect of SCT.

Reply 9. Representative images of the ARC of SCTR-Cre::ROSA-tdTomato double transgenic mice. Scale bars = 200 μ m.

References

1 Liu Y, et al. A gut-brain axis mediates sodium appetite via gastrointestinal peptide regulation on a medulla-hypothalamic circuit. *Sci Adv.* 2023 Feb 15;9(7).

2 Sternson SM, et al. Topographic mapping of VMH  arcuate nucleus microcircuits and their reorganization by fasting. *Nat Neurosci.* 2005 Oct;8(10):1356-63. doi: 10.1038/nn1550.

3 Rau AR, et al. Energy state alters regulation of proopiomelanocortin neurons by glutamatergic ventromedial hypothalamus neurons: pre- and postsynaptic mechanisms. *J Neurophysiol.* 2021 Mar 1;125(3):720-730. doi: 10.1152/jn.00359.2020.

4 Sun JS, et al. Ventromedial hypothalamic primary cilia control energy and skeletal homeostasis. *J Clin Invest.* 2021 Jan 4;131(1):e138107. doi: 10.1172/JCI138107.

5 Moore AN, et al. Neuronal activity increases the phosphorylation of the transcription factor cAMP response element-binding protein (CREB) in rat hippocampus and cortex. *J Biol Chem.* 1996 Jun 14;271(24):14214-20. doi: 10.1074/jbc.271.24.14214.

6 Karki P, et al. cAMP response element-binding protein (CREB) and nuclear factor κ B mediate the tamoxifen-induced up-regulation of glutamate transporter 1 (GLT-1) in rat astrocytes. *J Biol Chem.* 2013 Oct 4;288(40):28975-86. doi: 10.1074/jbc.M113.483826.

The new data on food intake and body mass regulation in global SCT and SCTR knockout mice fed regular chow diet show genotype effects that were not seen in the previous work. This should be mentioned / discussed.

We appreciate the suggestion of the reviewer and have added a discussion related to this, please also see the Discussion, Line 386-399.

Sctr and Sct must be added to extended Figure 4C.

We have added Sctr and Sct to Supplementary Fig. 11.

Line 238: TH is an enzyme, not a NE precursor.

We have corrected this misnomer and revised this sentence to “..... tyrosine hydroxylase (TH), a rate-limiting enzyme in the biosynthesis of NE, were both significantly downregulated in iBAT of ShSCT mice compared with ShCon control.”, please also see Line 255.

Line 240: improve the wording: The impaired activity or function or tone (instead of innervation) of TH+ sympathetic nerves

We agree with the reviewers and this sentence has been revised to “The impaired function of TH+ sympathetic nerves in iBAT following the loss of VMH-derived SCT was further confirmed by immunofluorescent staining.”, please also see Line 257.

The Discussion should end with a paragraph summarizing the main findings and presenting a perspective.

We thank the reviewer for raising this point. We have added a concluding paragraph. Please see the Discussion, Line 440-446.

We sincerely thank the editor and reviewers for your time and invaluable insights, which have helped us to significantly improve and strengthen our current work.

On behalf of all authors,
Billy K.C. Chow

REVIEWERS' COMMENTS

Reviewer #1 (Remarks to the Author):

The authors have provide additional data and information which satisfies my concerns

Reviewer #2 (Remarks to the Author):

The authors have satisfactorily addressed my questions.

Reviewer #3 (Remarks to the Author):

All points of criticism raised by reviewer 3 were thoroughly addressed by conducting additional experiments. Related to the quantitation of UCP1 levels in BAT, the application of ELISA is somewhat unusual in the field. One my want to see a validation of these data by standard Western blotting.

Reviewer #1:

Reviewer #1 (Remarks to the Author):

The authors have provided additional data and information which satisfies my concerns

We thank the reviewer for their time and effort in the review of our manuscript.

Reviewer #2 (Remarks to the Author):

The authors have satisfactorily addressed my questions.

We thank the reviewer for their time and effort in the review of our manuscript.

Reviewer #3 (Remarks to the Author):

All points of criticism raised by reviewer 3 were thoroughly addressed by conducting additional experiments. Related to the quantitation of UCP1 levels in BAT, the application of ELISA is somewhat unusual in the field. One my want to see a validation of these data by standard Western blotting.

We thank the reviewer again for their kind comments and suggestions. We used the ELISA method with the aim of addressing reviewer 1's question regarding the absolute content of Ucp1 in BAT. Although less common, this application has also been reported¹. Here, we agree with the reviewer's suggestion and verified the ELISA results using Western blotting in the revised manuscript. Consistently, WB results showed that the level of Ucp1 was down-regulated in iBAT of ShSCT mice (see below Reply 1 or Supplementary Fig. 6m).

Reply 1. Western blot of Ucp1 in iBAT of ShCon and ShSCT mice. Numbers in parentheses in each graph indicate sample size. Box plots with whiskers from minima to maxima, the central line at the 50th percentile, and the ends of the box at the 25th and 75th percentiles. Two-tailed Student's *t*-test. ****P* < 0.001. Error bars represent SEM. Source data are provided as a Source Data file.

1 Whitehead A, et al. Brown and beige adipose tissue regulate systemic metabolism through a metabolite interorgan signaling axis. Nat Commun. 2021 26;12(1):1905.

We sincerely thank the editor and reviewers for your time and invaluable insights, which have helped us to significantly improve and strengthen our current work.

On behalf of all authors,
Billy K.C. Chow